# Single-cell transcriptomics of human T cells reveals tissue and activation signatures in health and disease

Peter A. Szabo [1,6], Hanna Mendes Levitin[2,6], Michelle Miron[1,3], Mark E. Snyder[1], Takashi Senda[1,4], Jinzhou Yuan[2], Yim Ling Cheng[2], Erin C. Bush[2], Pranay Dogra[1], Puspa Thapa[1], Donna L. Farber [1,3,4,7]* & Peter A. Sims [2,5,7]*

Human T cells coordinate adaptive immunity in diverse anatomic compartments through production of cytokines and effector molecules, but it is unclear how tissue site influences T cell persistence and function. Here, we use single cell RNA-sequencing (scRNA-seq) to define the heterogeneity of human T cells isolated from lungs, lymph nodes, bone marrow and blood, and their functional responses following stimulation. Through analysis of >50,000 resting and activated T cells, we reveal tissue T cell signatures in mucosal and lymphoid sites, and lineage-specific activation states across all sites including distinct effector states for CD8[+] T cells and an interferon-response state for CD4[+] T cells. Comparing scRNA-seq profiles of tumor-associated T cells to our dataset reveals predominant activated CD8[+] compared to CD4[+] T cell states within multiple tumor types. Our results therefore establish a high dimensional reference map of human T cell activation in health for analyzing T cells in disease.

[1] Columbia Center for Translational Immunology, Columbia University Irving Medical Center, New York, NY, USA. [2] Department of Systems Biology, Columbia University Irving Medical Center, New York, NY, USA. [3] Department of Microbiology and Immunology, Columbia University Irving Medical Center, New York, NY, USA. [4] Department of Surgery, Columbia University Irving Medical Center, New York, NY, USA. [5] Department of Biochemistry and Molecular Biophysics, Columbia University Irving Medical Center, New York, NY, USA. [6] These authors contributed equally: Peter A. Szabo, Hanna Mendes Levitin. [7] These authors jointly supervised this work: Donna L. Farber, Peter A. Sims. *email: df2396@cumc.columbia.edu; pas2182@cumc.columbia.edu

T lymphocytes coordinate adaptive immune responses and are essential for establishing protective immunity and maintaining immune homeostasis. Activation of naive T cells through the antigen-specific T cell receptor (TCR) initiates transcriptional programs that drive differentiation of lineage-specific effector functions; CD4[+] T cells secrete cytokines to recruit and activate other immune cells while CD8[+] T cells acquire cytotoxic functions to directly kill infected or tumor cells. Most of these effector cells are short-lived, although some develop into long-lived memory T cells which persist as circulating central (TCM) and effector-memory (TEM) subsets, and non-circulating tissue resident memory T cells (TRM) in diverse lymphoid and non-lymphoid sites[1–4]. Recent studies in mouse models have established an important role for CD4[+] and CD8[+] TRM in mediating protective immunity to diverse pathogens[2,5–7]. Defining how tissue site impacts T cell function is therefore important for targeting T cell immunity.

In humans, most of our knowledge of T cell activation and function derives from the sampling of peripheral blood. Recent studies in human tissues have revealed that the majority of human T cells are localized in lymphoid, mucosal and barrier tissues[8] and that T cell subset composition is a function of the specific tissue site[9,10]. Human TRM cells can be defined based on their phenotypic homology to mouse TRM and are distinguished from circulating T cells in blood and tissues by a core transcriptional and protein signature[10–13]. However, the role of tissue site in determining T cell functional responses, and a deeper understanding of the relationship between blood and tissue T cells beyond composition differences are key unanswered questions in human immunology.

The functional responses of T cells following antigen or pathogen exposure have been largely defined in mouse models, and are generally classified based on whether or not they secrete specific cytokines or effector molecules. Effector CD4[+] T cells comprise different functional subtypes (Th1 cells secrete IFN-γ and IL-2; Th2 secrete IL-4, 13; Th17 secrete IL-17A, etc.)[14], while effector CD8[+] T cells secrete pro-inflammatory cytokines (IFN-γ, TNF-α) and/or cytotoxic mediators (perforin and granzymes)[15]. Certain conditions can lead to inhibition of functional responses; for example, CD4[+] T cells encountering self-antigen become anergic and fail to produce IL-2, while CD8[+] T cells responding to chronic infection, tumors, or lacking CD4[+] T cell help become functionally exhausted, and express multiple inhibitory molecules (e.g., PD-1, LAG3)[16–18]. While human T cells can produce similar cytokines, effector and inhibitory molecules as mouse counterparts[19–22], the full complement of functional responses for human T cells in tissues has not been elucidated. Thus, establishing a baseline of healthy T cell states in humans is essential for defining dysregulated and pathological functions of T cells in disease.

Single cell transcriptome profiling (scRNA-seq) has enabled high resolution mapping of cellular heterogeneity, development, and activation states in diverse systems[23,24]. This approach has been applied to analyze human T cells in diseased tissues[25,26] and in response to immunotherapies in cancer[27]; however, baseline functional profiles of human T cells in healthy blood and tissues would be an important reference dataset. We have established a tissue resource where we obtain multiple lymphoid, mucosal, and other peripheral tissue sites from human organ donors[9–11,13,28,29], enabling study of T cells across different anatomical spaces.

Here, we used scRNA-seq of over 50,000 resting and activated T cells from lung (LG), lymph nodes (LN), bone marrow (BM) and blood, along with integrated computational analysis to define cellular states of homeostasis and activation of human blood and tissue-derived T cells. We reveal how human T cells in tissues relate to those in blood, and identify a conserved tissue signature and activation states for human CD4[+] and CD8[+] T cells conserved across all sites. We further show how scRNA-seq profiles of T cells associated with human tumors can be projected onto this healthy baseline dataset, revealing their functional state. Our results establish a high dimensional reference map of human T cell homeostasis and function in multiple sites, from which to define the origin, composition and function of T cells in disease.

## Results

**scRNA-seq analysis of human T cells in blood and tissues.** We obtained BM, LN, and LG as representative primary lymphoid, secondary lymphoid and mucosal tissue sites, respectively, from two deceased adult organ donors who met the criteria of health for donation of physiologically healthy tissues for lifesaving transplantation, being free of chronic disease and cancer (Supplementary Table 1). For comparison, we obtained blood from two healthy adult volunteers. CD3[+] T cells isolated from tissues and blood were cultured in media alone ("resting") or in the presence of anti-CD3/anti-CD28 antibodies ("activated") (Fig. 1a). Single cells were encapsulated for cDNA synthesis and barcoded using the 10x Genomics Chromium system, followed by library construction, sequencing, and computational identification of T cells (Supplementary Fig. 1, Supplementary Table 2, Supplementary Data 1).

We initially analyzed tissue T cell populations from the two individual donors, comprising six samples per donor (resting and activated samples from three tissue sites). We merged all data for each donor, performed unsupervised community detection[30] to cluster the data based on highly variable genes (Supplementary Data 2), and projected cells in two dimensions using Uniform Manifold Approximation and Projection (UMAP)[31]. For both donors, the dominant sources of variation between cells were activation state (vertical axis) and CD4/CD8 lineage (horizontal axis) (Fig. 1b). Tissue site was also a source of variability; T cells from BM and LN co-localized while LG T cells were more distinct (Fig. 1b), consistent with a greater proportion of CD8[+] T cells and TRM phenotype cells in LG relative to the two lymphoid sites (Supplementary Fig. 2 and previous studies[10,13,32]).

Differential gene expression from the scRNA-seq data resolved T cell subsets and functional states within and between sites and lineages into 10–11 clusters (Fig. 1c, Supplementary Data 3, 4). CD4[+] T cells comprised 6–7 clusters: resting cells expressing *CCR7*, *SELL* and *TCF7*, (corresponding to naive or TCM cells); three activation-associated clusters expressing *IL2*, *TNF*, and *IL4R* at different levels; TRM-like resting and activated clusters expressing canonical TRM markers *CXCR6* and *ITGA1*[13,33]; and a distinct regulatory T cell (Treg) cluster expressing Treg-defining genes *FOXP3*, *IL2RA*, and *CTLA4* (Fig. 1c). CD8[+] T cells comprised four clusters distinct from CD4[+] T cells and included: two TEM/TRM-like clusters expressing *CCL5*, cytotoxicity-associated genes (*GZMB*, *GZMK*), and TRM markers (*CXCR6*, *ITGA1*); an activated TRM/TEM cluster expressing *IFNG*, *CCL4*, *CCL3*; and clusters representing terminally differentiated effector cells (TEMRA) expressing cytotoxic markers *PRF1* and *NKG7* (Fig. 1c). In terms of tissue distribution, TRM cells were largely in the lung, Tregs were primarily identified in LN, while TEMRA cells were enriched in BM (consistent with phenotype analysis, Supplementary Fig. 2); the remaining resting and activated CD4[+] and CD8[+] T cell clusters derived from all sites (Fig. 1b, c). These results show subset-specific profiles in human tissues, but suggest similar activation profiles across sites.

To assess how blood T cells relate to those in tissue, we performed scRNA-seq analysis of resting and activated blood T cells from two adult donors, and projected the merged data

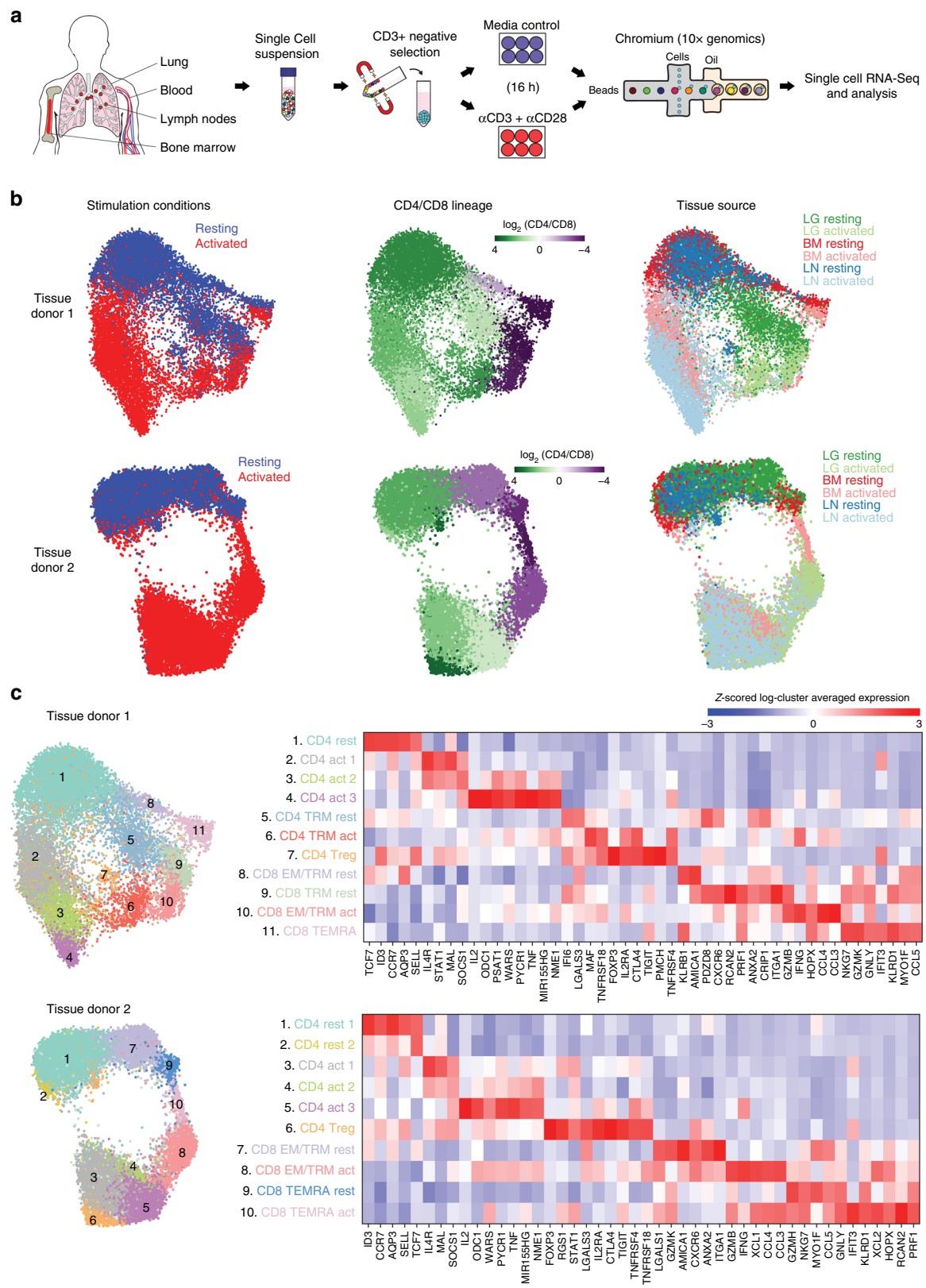

onto the UMAP embeddings of T cells from each tissue donor (Fig. 2a, b). The majority of blood T cells co-localized with resting or activated T cells from BM but did not exhibit substantial overlap with LG or LN T cells from either donor, particularly in the resting state (Fig. 2a, b). We also quantified the number of blood T cells that were transcriptionally similar to CD4+ and CD8+ T cells from each tissue within resting or activated samples (Fig. 2c, d). Resting blood T cells were highly represented among CD4+ and CD8+ T cells in BM (Fig. 2c, d). Interestingly, a substantial number of unstimulated blood T cells projected onto

**Fig. 1** Single-cell RNA-seq analysis of resting and activated T cells from multiple tissue sites. **a** Experimental workflow for single-cell analysis of T cells from human tissues and blood including magnetic negative selection of CD3[+] cells, in vitro culture and activation, and Chromium 3′-scRNA-seq. **b** UMAP embeddings of merged scRNA-seq profiles from resting and activated T cells from lung (LG), bone marrow (BM), and lung-draining lymph node (LN) in each of two organ donors colored by resting/activated condition, CD4/CD8 expression ratio (all cells in a given cluster assigned the same average value), and tissue source. **c** Identification of T cell subpopulations. UMAP embeddings colored by expression cluster along with heatmaps showing z-scored average expression of curated T cell subset marker genes that had a fold change >2 and $p < 0.05$ by the binomial test for at least one cluster. Genes are ordered by the cluster in which they have the highest enrichment. Subsets designated based on resting ("rest") or activated ("act") condition and expression of known markers denoting effector memory (TEM), tissue resident memory (TRM), terminally differentiated effector cells (TEMRA), and regulatory T cells (Treg). Source data for **c** detailing averaged expression values for T cell subset marker genes are provided in the Source Data file

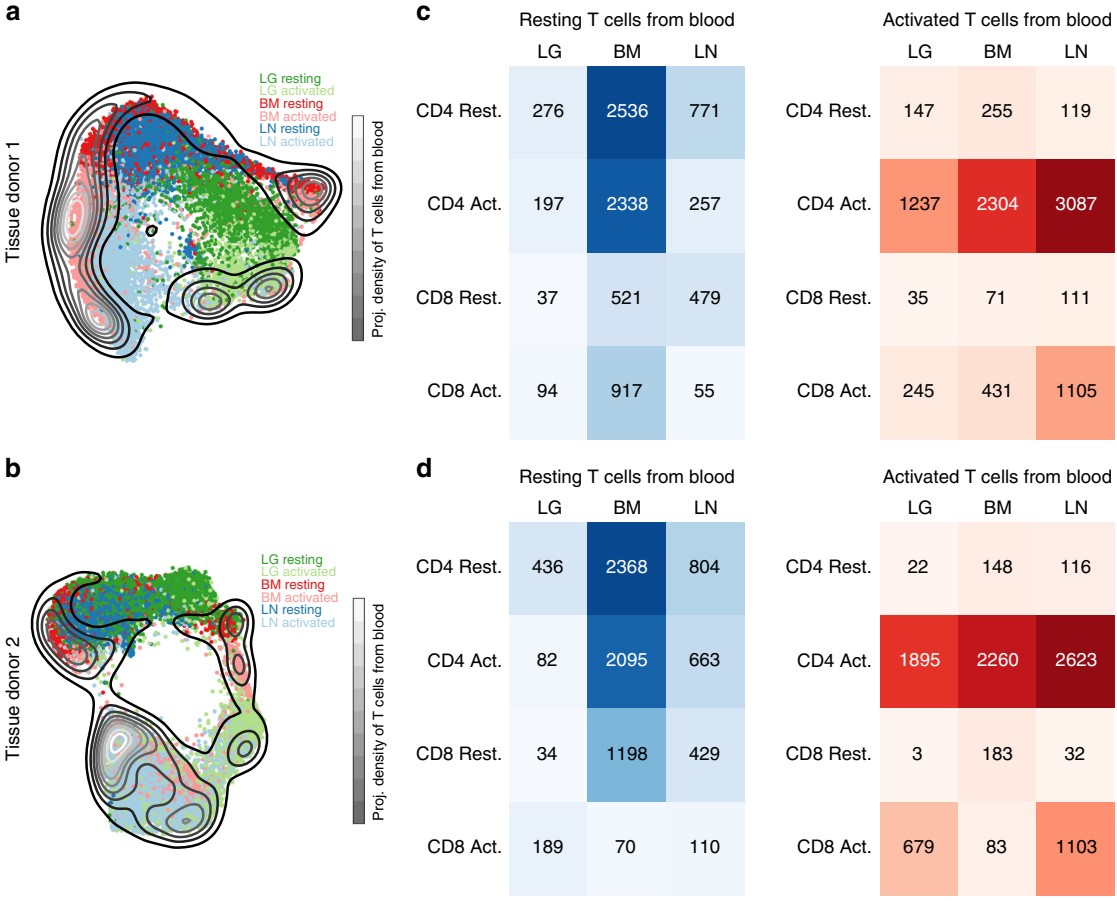

**Fig. 2** Comparison of blood and tissue T cells. **a** UMAP embedding of T cells from tissue donor 1 colored by tissue and overlaid with a contour plot corresponding to the UMAP projection of the combined resting and activated T cells from two blood donors onto the tissue embedding. **b** Same as (**a**) for organ donor 2. **c** Heatmaps showing the number of blood T cells that project most closely to each tissue/stimulation status combination in the tissue donor 1 UMAP embedding. **d** Same as (**c**) for tissue donor 2

activated CD4[+] T cells in BM for both donors (Fig. 2c, d, left panels). In contrast, activated blood T cells were strongly represented among activated CD4[+] T cells for all tissue sites and in LN for CD8[+] T cells (Fig. 2c, d; right panels). Similar results were obtained when each blood sample was compared separately to each tissue donor (Supplementary Fig. 3), and when blood T cells were projected onto tissue T cells using *scmap*[34], an alternative scRNA-seq data projection package (Supplementary Fig. 4). These results indicate that resting blood T cells are most similar to those in the BM, while activated blood and tissue-derived T cells share common signatures.

**Identifying a tissue gene signature in multiple sites**. The major transcriptional differences between tissue and blood T cells based on population-level RNAseq originate from the presence of TRM

in tissues[13]. Because scRNA-seq enables high-resolution detection of gene expression differences that can be unambiguously traced to individual T cells, we investigated whether there were intrinsic features of tissue T cells that distinguished them from blood. Resting memory T cells in tissues and blood express high levels of CCL5 (Supplementary Fig. 5), a marker of CD8[+] TEM cells[35], enabling direct comparison of gene expression between similar subsets. We identified a similar complement of genes that were highly expressed in TEM cells from each tissue compared to blood (Fig. 3a–c). Interestingly, these tissue-intrinsic genes include those associated with microtubules and cytoskeleton (tubulin-encoding genes *TUBA1A, TUBA1B, TUBB, TUBB4B; S100A4*) as well as genes encoding cell matrix, membrane scaffolding, and adhesion molecules (*VIM* or vimentin, galectins *LGALS1/LGALS3, AMICA1, ITM2C, EZR*, annexins *ANXA1/ ANXA2*) (Fig. 3a–c). TRM signature genes including ITGA1 and

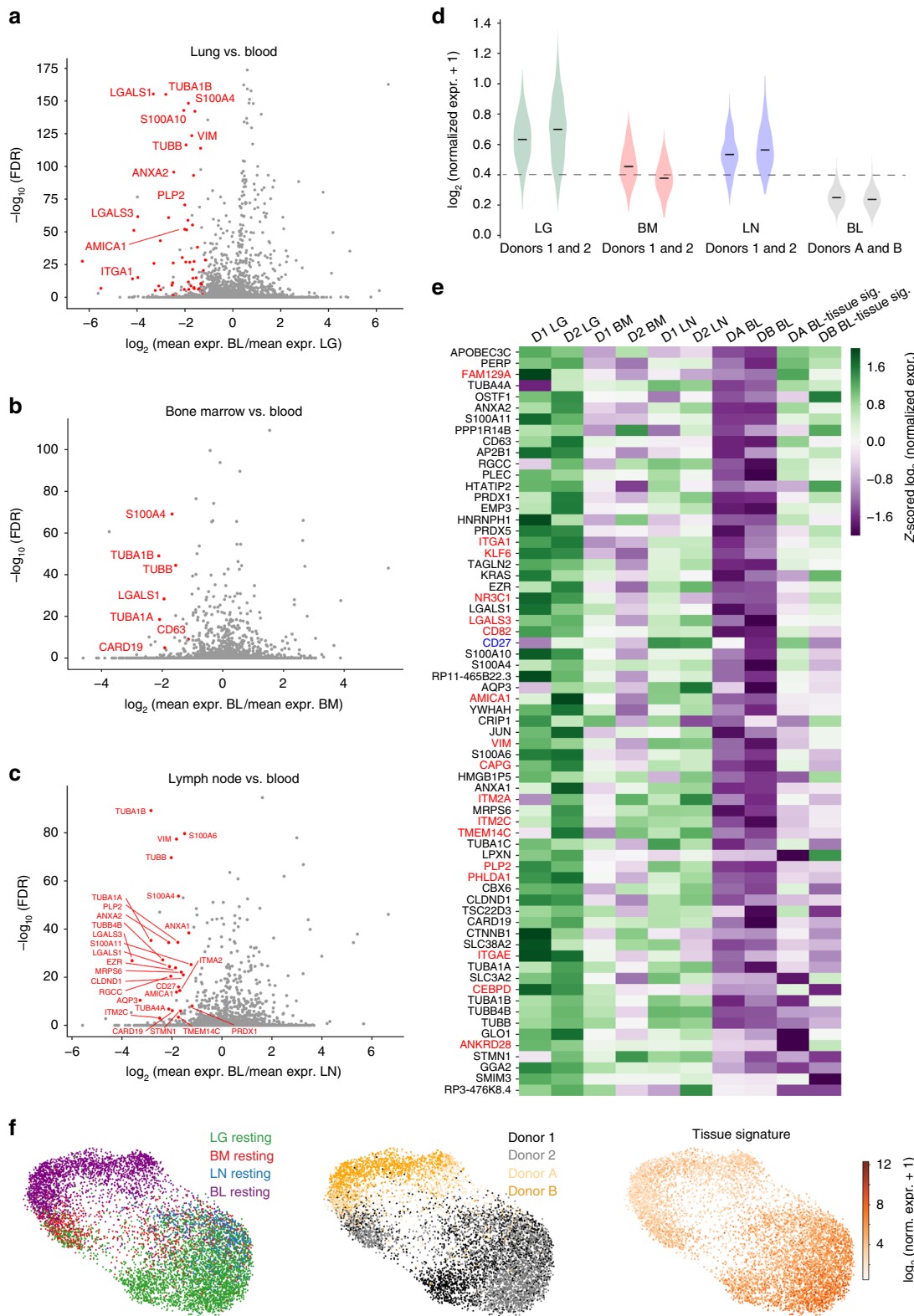

ITGAE were also upregulated in tissues compared to blood, particularly in the lung (Fig. 3a–c). These findings suggest that localization of T cells in tissues likely involves structural changes in the cell that facilitate interactions with tissue matrix.

We next compared the single-cell distribution of average expression of tissue signature genes in the blood and tissues (Fig. 3d). CCL5$^+$ TEM cells from all three tissues (both donors) express higher levels of tissue signature genes compared to blood,

though LG and LN T cells have higher expression than those from BM (Fig. 3d). Notably, a minute fraction of blood TEM cells (<0.5%) express this tissue signature at levels comparable to that in LN (within one standard deviation of the mean for all tissues). Shown in a heat map are the relative expression levels for genes within the tissue signature, including genes enriched in human TRM cells[13,36], and genes associated with cytoskeletal, cell-matrix interactions, cell division, apoptotis, and signaling (Fig. 3e).

**Fig. 3** Identification of a tissue gene signature for resting memory T cells. **a** Volcano plot showing the average log-fold-change and average Benjamini-Hochberg-corrected *p*-values (FDR) for pairwise differential expression between CCL5+ T cells from each resting LG sample and each resting blood sample. Genes with negative log-fold-change are more highly expressed among CCL5+ cells in LG, with several differentially expressed genes (multiple test-corrected Wilcoxon *p* < 0.05, fold change >2) highlighted in red. **b** Same as (**a**) for comparison of resting CCL5+ T cells in BM and blood. **c** Same as (**a**) for comparison of resting CCL5+ T cells in LN in blood. **d** Violin plot showing the distributions of the average expression of all genes with two-fold higher expression (on average) in any tissue compared to blood and average FDR < 0.05 (described above) in any tissue for the resting CCL5+ T cells in each tissue and blood sample. The dashed line marks one standard deviation below the mean for average expression of this signature for all tissues (note a small number of blood cells fall above this line). **e** Heatmap shows z-scored average expression for all genes in the tissue signature from (**d**) among the resting CCL5+ T cells from each tissue and blood sample plus that of the rare blood subpopulation from (**d**), which expresses high levels of a subset of tissue signature genes. Previously identified TRM-associated genes from bulk RNA-seq studies are highlighted in red (enriched in CD69+ vs. CD69− tissue memory T cells)[13], and CD27 highlighted in blue was previously found to be upregulated on human TRM compared to TEM cells[36]. **f** UMAP embedding of resting *CCL5*+ T cells (TEM cells) from all four donors generated using the tissue-associated T cell signature colored by tissue site (left), donor (center), and average expression of the signature (right). Source data listing genes and expression values for (**a**–**e**) are provided in the Source Data file

Expression of the tissue signature genes is highest in LG, followed by LN and BM expressing only a subset of tissue-associated genes; the outlier subpopulation from blood expresses a fraction (<40%) of tissue signature genes at levels comparable to those in tissues (Fig. 3e). When resting TEM cells from all sites and donors were visualized by UMAP using the tissue-associated signature genes, blood T cells clustered distinctly from all tissues, while LG T cells clustered distinctly from LN and BM (Fig. 3f). Notably, a subset of T cells from BM and LG clustered more closely to blood T cells (Fig. 3f), indicating the presence of circulating T cells within these sites. Together, these results show that tissue T cells express genes associated with infiltration and localization in tissues along with residency markers, while blood contains only trace numbers of cells expressing these genes.

The tissue signature identified in Fig. 3 compared tissue from deceased organ donors to blood from living individuals. To establish that the observed differences were not due to tissue processing and/or T cells from organ donors versus living individuals, we analyzed our tissue signature in scRNAseq data from several available datasets, including BM from living individuals[37] and additional blood data (see Methods). We found that the tissue signature was significantly enriched in BM from living individuals, compared to blood (Supplementary Fig. 6a). Similarly, we found the tissue signature was enriched in all organ donor sites compared to blood from additional living donors (Supplementary Fig. 6b). Together, these results indicate that the tissue signature is an intrinsic feature of T cells from non-blood sites, and that our results from blood and BM are representative of T cells in these sites and representative of diverse individuals.

**Activation-induced transcriptional states across sites**. The clustering analysis in Fig. 2 suggested that activated T cells were more similar across sites than resting counterparts. To uncover gene expression patterns that were conserved across T cell populations in different tissues throughout activation, we applied a new analytical method called single-cell Hierarchical Poisson Factorization (scHPF)[38]. The scHPF algorithm identifies a small number of expression patterns, called factors that vary coherently across cells. These factors can represent discrete, subpopulation-specific programs or continuous programs like T cell activation that are expressed as a gradient across cells in different stages of a biological process. We applied scHPF to merged resting and activated T cells from each tissue and donor separately and hierarchically clustered the resulting factors (Fig. 4a, Supplementary Figs. 7, 8a). This analysis revealed seven gene expression modules (three resting and four activated/functional) that were highly conserved across tissues and donors, for which the highest scoring genes formed interpretable gene signatures (Fig. 4a, Supplementary Fig. 8a, Supplementary Data 5). Modules were annotated based on known markers among their highest scoring

genes, association with resting or activated states, and CD4:CD8 experssion ratio. The three modules associated with a resting state (Fig. 4a) included a Treg module defined by canonical genes (*FOXP3, CTLA4, IRF4, TNFRSF4* (OX40)[39]); a putative resting CD4+ Naive/Central memory (NV/CM) module enriched in CD4+ T cells and defined by genes associated with lymphoid homing, egress and quiescence (*SELL, KLF2, LEF1*, respectively); and a CD4+/CD8+ Resting module, distinguished by expression of *IL7R*, a receptor required for T cell survival[40,41], and *AQP3*, which encodes a water channel protein of unclear function in lymphocytes[42]. Importantly, the CD4+/CD8+ Resting module did not contain factors from blood and had the highest enrichment for the tissue signature identified in Fig. 3 (Supplementary Fig. 9).

Four modules were associated with T cell activation and/or function, some of which were lineage-specific. A Proliferation module expressed by activated CD4+ and CD8+ lineages included genes associated with T cell activation/proliferation (*IL2, LIF*) and cell division (*CENPV, G0S2, ORC6*) (Fig. 4a). This module was also marked by expression of *NME1*, a metastasis suppressor/endonuclease-encoding gene[43] not previously associated with T cells (Fig. 4a). An Interferon (IFN) Response module enriched among activated CD4+ T cells included multiple gene families associated with canonical IFN responses[44–46] (*IFIT3, IFIT2, STAT1, MX1, IRF7*, and *JAK2*). In contrast, CD8+ T cell-enriched modules included a Cytotoxic module, containing genes associated with cytotoxicity (*GNLY, GZMK*) and transcription factors associated with effector/memory differentiation (*ZEB2, EOMES, ZNF683*)[46–48], and a Cytokine module with genes encoding chemokines and cytokines (*CCL3, CCL4, CCL20, IFNG, IL10, TNF*), inhibitory molecules (*LAG3, CD226* (TIGIT), *HAVCR2* (TIM3)), and the widely expressed homeobox protein *HOPX*[49]. These results indicate a limited spectrum of functional states for human T cells across blood and tissue sites.

To understand how these gene modules correspond to resting and activated states in CD4+ and CD8+ T cells, we visualized the average expression of their top-ranked genes on diffusion maps for each donor and tissue (Fig. 4b–e). This visualization defined activation trajectories with resting T cells on the left (blue) and activated T cells projecting to the right (red; Fig. 4b, c). In all four sites in both individuals, module expression for CD4+ T cell was positioned along activation trajectories from CD4 NV/CM Resting (left) to IFN-Response (middle) to Proliferation (right) (Fig. 4d). Expression of genes within the Proliferation module co-localized with peak expression of *NME1* and *IL2RA* (Supplementary Fig. 8b, c), while the IFN Response module genes exhibited peak expression at the middle of the trajectory as exemplified by *IFIT3* expression (top ranked gene) (Supplementary Fig. 8d), suggesting a potential intermediate activation state.

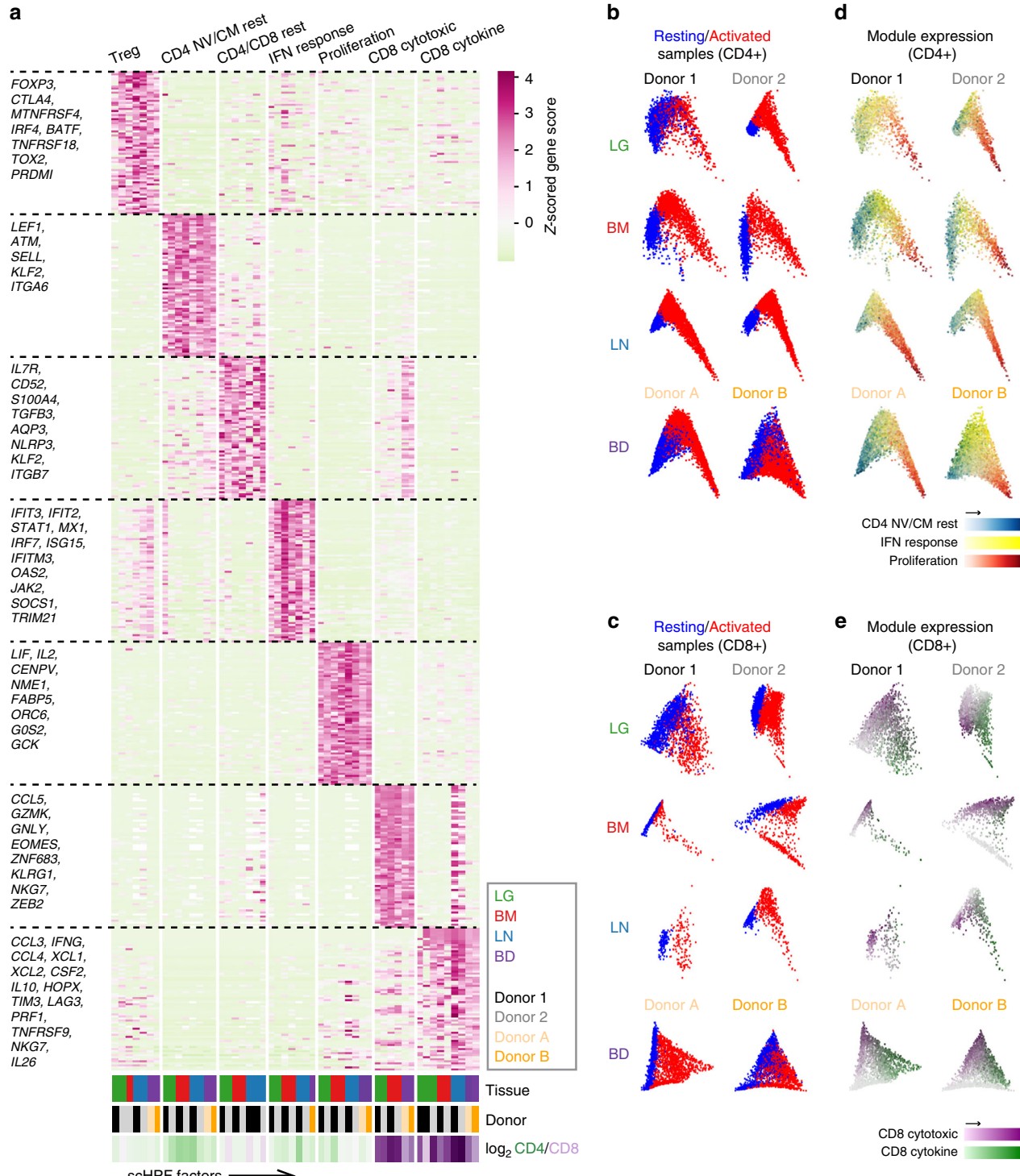

**Fig. 4** Defining conserved transcriptional states in resting and activated T cells by single-cell Hierarchical Poisson Factorization (scHPF). **a** Heatmap shows gene scores for the top genes (rows) in each expression module identified by clustering scHPF factors (columns) that were computed in separate analyses of cells from each tissue and donor (Supplementary Fig. 8a). Selected genes are indicated to the left, and complete lists of top genes are available in Supplementary Data 5. Color bars at the bottom of the heatmap indicate each factors' tissue of origin, donor of origin, and CD4/CD8 bias. (NV/CM = naive or TCM). **b** Diffusion maps of CD4+ T cells in each tissue and donor, with cells colored by sample origin as resting (blue) or activated (red). **c** Same as (**b**) but for CD8+ T cells. **d** Diffusion maps of CD4+ T cells from (**b**), with cells colored by their average expression of the top genes from scHPF expression modules. Colors for different modules (CD4 NV/CM Resting, IFN Response, Proliferation) were blended using the RGB color model. **e** Diffusion maps of CD8+ T cells from (**c**), with cells colored by their average expression of the top genes from scHPF expression modules (CD8 Cytotoxic and CD8 Cytokine). Source data listing gene scores in (**a**) are provided in the Source Data file

In CD8$^+$ T cells, the Cytokine module localized in the most activated cells for all sites also shown by *IFNG* expression (Fig. 4e, Supplementary Fig. 8e), while the Cytotoxic module was expressed among resting and activated cells (Fig. 4e). Therefore, scHPF takes an unbiased approach to uncover major functional states, reference signatures and activation trajectories for human T cells that are conserved across sites.

**A type II IFN response state in activated CD4$^+$ T cells**. The functional states identified for human CD8$^+$ T cells in Fig. 4 were consistent in with those seen in vivo in mouse infection models[15]. By contrast, the modules identified for CD4$^+$ T cell activation revealed markers and functional states not typically associated with effector CD4$^+$ T cells. We therefore assessed expression kinetics of the top-scoring genes in the Proliferation and IFN Response modules, *NME1* and *IFIT3*, respectively, during the course of T cell activation ex vivo by qPCR. Expression of *NME1* transcripts rapidly increased after TCR-stimulation, peaking between 16 and 24 h and remaining elevated for up to 72 h, for both CD4$^+$ and CD8$^+$ T cells compared to unstimulated controls, a pattern of expression similar to the canonical T cell activation marker *IL2RA* (Fig. 5a). Notably, the extent of activation-associated upregulation of *NME1* transcripts was greater in CD4$^+$ compared to CD8$^+$ T cells, while *IL2RA* was more upregulated in CD8$^+$ T cells (Fig. 5a). At the protein level, NME1 expression increased in CD4$^+$ and CD8$^+$ T cells after TCR-mediated stimulation from 24 to 120 h (Fig. 5b, upper), and with each successive round of T cell proliferation, while CD25 was expressed similarly, independent of cell division (Fig. 5b, lower). These results establish NME1 expression as a marker of T cell activation, coupled to the extent of proliferation.

In contrast to *NME1*/*IL2RA* upregulation, expression of the interferon-inducible transcript *IFIT3*, showed transient upregulation by CD4$^+$ T cells following TCR-stimulation, peaking at 16 h and returning to near baseline levels by 48 h post-stimulation (Fig. 5c). By contrast, induction of *IFIT3* by culturing T cells with IFN-α (type I) or IFN-γ (type II) occurred rapidly (within 2 h) and persisted throughout the culture period (Fig. 5d, e). To identify the contribution of type I or type II IFN signaling to TCR-triggered *IFIT3* induction, we included blocking antibodies to type I or Type II IFN in the cultures. While neutralizing antibodies for type I IFNs and IFNαR2 completely inhibited *IFIT3* induction by Type I IFN, TCR-mediated upregulation of *IFIT3* was unaffected (Fig. 5d). However, blockade of type II IFN signaling via a combination of anti-IFNγ and anti-IFNγR1 antibodies inhibited upregulation of *IFIT3* by both exogenous IFN-γ and TCR-mediated stimulation (Fig. 5e). Importantly, blocking type II (or type I) IFN signaling did not inhibit T cell activation as assessed by induction of *NME1* transcript expression, and addition of IFN-α or −γ did not induce *NME1* expression (Fig. 5d, e). These results establish that the IFN-responsive state suggested by the scRNA-seq trajectories is recapitulated in real-time as part of an intermediate activation state driven by TCR-triggered IFN-γ production.

We further assessed whether CD4$^+$ T cells express *IFIT3* and *NME1* in vivo using a published scRNA-seq dataset of T cells isolated from the blood of dengue virus-infected patients, which contains a fraction of activated CD4$^+$ T cells[50]. Both *NME1* and *IFIT3* were expressed by CD4$^+$ T cells from dengue-infected patients (Supplementary Fig. 10). These results show that genes associated with functional modules identified for CD4$^+$ T cell activation are expressed in vivo.

**Defining functional states in tumor-associated T cells**. Although there have been several large-scale scRNA-seq studies

of disease-associated T cells, these data are generally not placed in the context of T cell activation in healthy individuals. To demonstrate the utility of our resource as a reference point for human disease, we used UMAP to project recently reported scRNA-seq profiles of tumor-associated T cells from four different human cancers onto our map of T cell activation states. We merged all of our T cell data from four donors and four sites in a single UMAP embedding (Fig. 6a), colored by tissue site, donor, stimulation, cluster-level CD4/CD8 status, and CCL5 expression, indicative of effector status. We projected scRNA-seq profiles of tumor-associated T cells from four different human cancers[27,51–53] (non-small cell lung cancer (NSCLC), colorectal cancer (CRC), breast cancer (BC), and melanoma (MEL)) onto this embedding to compare each tumor-associated T cell to healthy T cells (Fig. 6b, c). We also investigated expression of activation state and lineage markers in the healthy T cell embedding and tumor projections (Fig. 6c). Tumor-associated CD8$^+$ T cells project onto healthy CD8$^+$ T cells from all sites in both resting and activated states (Fig. 6b). Moreover, genes associated with TRM (CXCR6) and the Cytotoxic and Cytokine modules are all represented among tumor-associated CD8$^+$ T cells (Figs. 6c, 7). By contrast, tumor-associated CD4$^+$ T cells projected mostly onto resting blood and tissue T cells (Fig. 6b), while CD4$^+$ T cell activation states and associated markers (*NME1*, IFIT3) were largely absent (Fig. 6c). Projecting tumor-associated T cells onto each individual tissue and blood donor yielded results consistent with projection onto the combined dataset (Supplementary Figs. 11–14). We note that projecting tumor-associated T cells onto our reference map using the alternate projection algoritham *scmap*[34] showed similar results (Supplementary Fig. 15). This analysis reveals that tumor-associated T cells contain activated CD8$^+$ T cell states, but lack the presence of functionally activated CD4$^+$ T cell states.

A hallmark of tumor-associated T cells is a state of hyporesponsiveness or functional exhaustion, marked by persistent expression of surface inhibitory markers including PD-1, CTLA4, LAG3, TIM3 and others, many of which are expressed following T cell activation[17,54,55]. Some of these molecules (PD-1, CTLA4) are important targets for immunotherapy to promote anti-tumor immunity[56–60]. We compared expression of exhaustion and functional markers across healthy and tumor-associated T cells (Fig. 7; Supplementary Figs. 16, 17). Tumor-associated CD8$^+$ T cells expressing exhaustion markers across all four tumor types project onto activated CD8$^+$ T cells in our map, and express genes within the Cytokine module (*CCL3, CCL4, XCL1, XCL2*, and *IFNG*), and to a lesser extent Cytotoxic module (Fig. 7; Supplementary Figs. 16, 17). Interestingly, a subset of these tumor-associated CD8$^+$ T cells, but not healthy T cells, express high levels of *MKI67*, associated with proliferating cells and other cell cycle control markers (Fig. 7, Supplementary Fig. 17). Therefore, tumor-associated T cells expressing exhaustion markers also express genes associated with normal CD8$^+$ effector T cell function and ongoing proliferation.

## Discussion
Human T cells persist in distinct anatomic sites, maintain protective immunity and surveillance, and are key targets for immune modulation in tumor immunotherapy, transplantation, and autoimmunity. Here, we used scRNA-seq profiling of resting and TCR-stimulated T cells from blood, lymphoid and mucosal tissues to generate a reference map of human T cells and understand how T cell homeostasis and function are related to the tissue site. Our findings demonstrate fundamental differences between T cells from tissues and blood, but similar functional and activation states across sites that are intrinsic to lineage; human CD4 T cell activation is defined by response to cytokines and

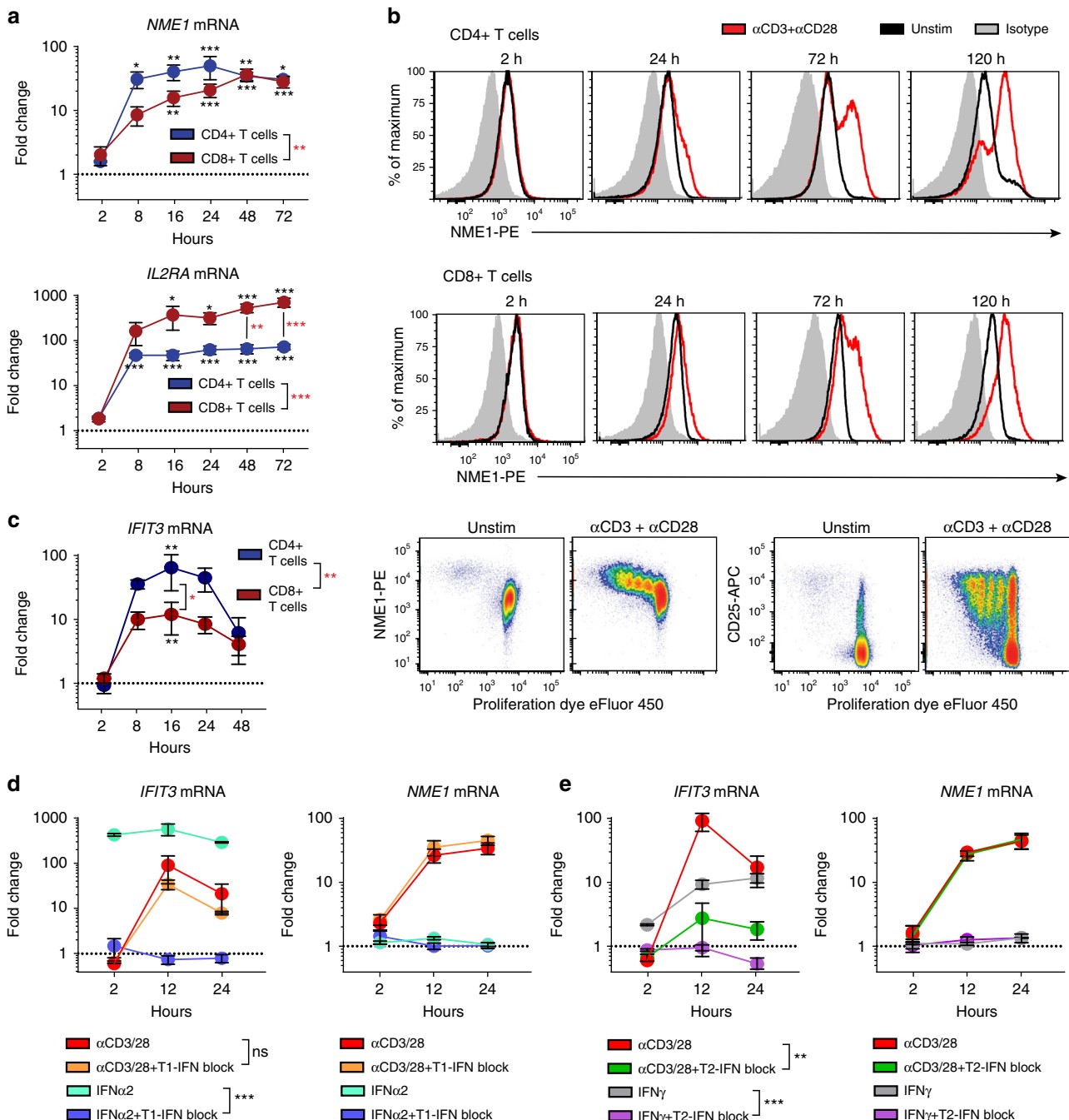

**Fig. 5** Induction of NME1 and IFIT3 expression during T cell activation. **a** Expression of *NME1* and *IL2RA* mRNA by blood CD4$^+$ or CD8$^+$ T cells after stimulation with anti-CD3/anti-CD28 antibodies by qPCR. Data shown as mean fold-change (±SEM) relative to unstimulated CD4$^+$ or CD8$^+$ T cell controls (dotted line) from 4 individuals (independent experiments). Statistical analysis between stimulated and unstimulated cells (black asterisk) or CD4$^+$ and CD8$^+$ T cells (red asterisk) made by two-way ANOVA with Sidak test for multiple comparisons. **b** Intracellular NME1 protein expression by blood T cells after stimulation for indicated timepoints (red) compared to unstimulated (black) and isotype control (gray). Bottom row: CD25 and NME1 expression by proliferating CD3$^+$ T cells after 5 days of stimulation. Data are representative of 4 individuals. **c** Expression of *IFIT3* mRNA in blood T cells by qPCR after TCR-stimulation, shown as mean fold-change (±SEM) relative to unstimulated controls (dotted line) for four individuals. Two-way ANOVA with Sidak test for multiple comparisons was used for statistical comparisons (black asterisk, stimulated versus unstimulated) or (red asterisk, CD4$^+$ versus CD8$^+$ T cells). **d** *IFIT3* or *NME1* mRNA expression in CD4$^+$ T cells after culture with anti-CD3/anti-CD28 or IFNα2 (1000 units/mL) ± type I IFN neutralizing antibody cocktail or **e** IFNγ (10 ng/mL) ± anti-IFNγ/anti-IFNγR1 antibodies (1 μg/mL each), shown as mean fold-change (±SEM) relative to unstimulated controls (dotted line) for three individuals. Statistical comparisons made by two-way ANOVA. For all panels: "ns" denotes not significant; *$p \leq 0.05$; **$p \leq 0.01$; ***$p \leq 0.001$. Source data for gene expression values are provided in the Source Data file

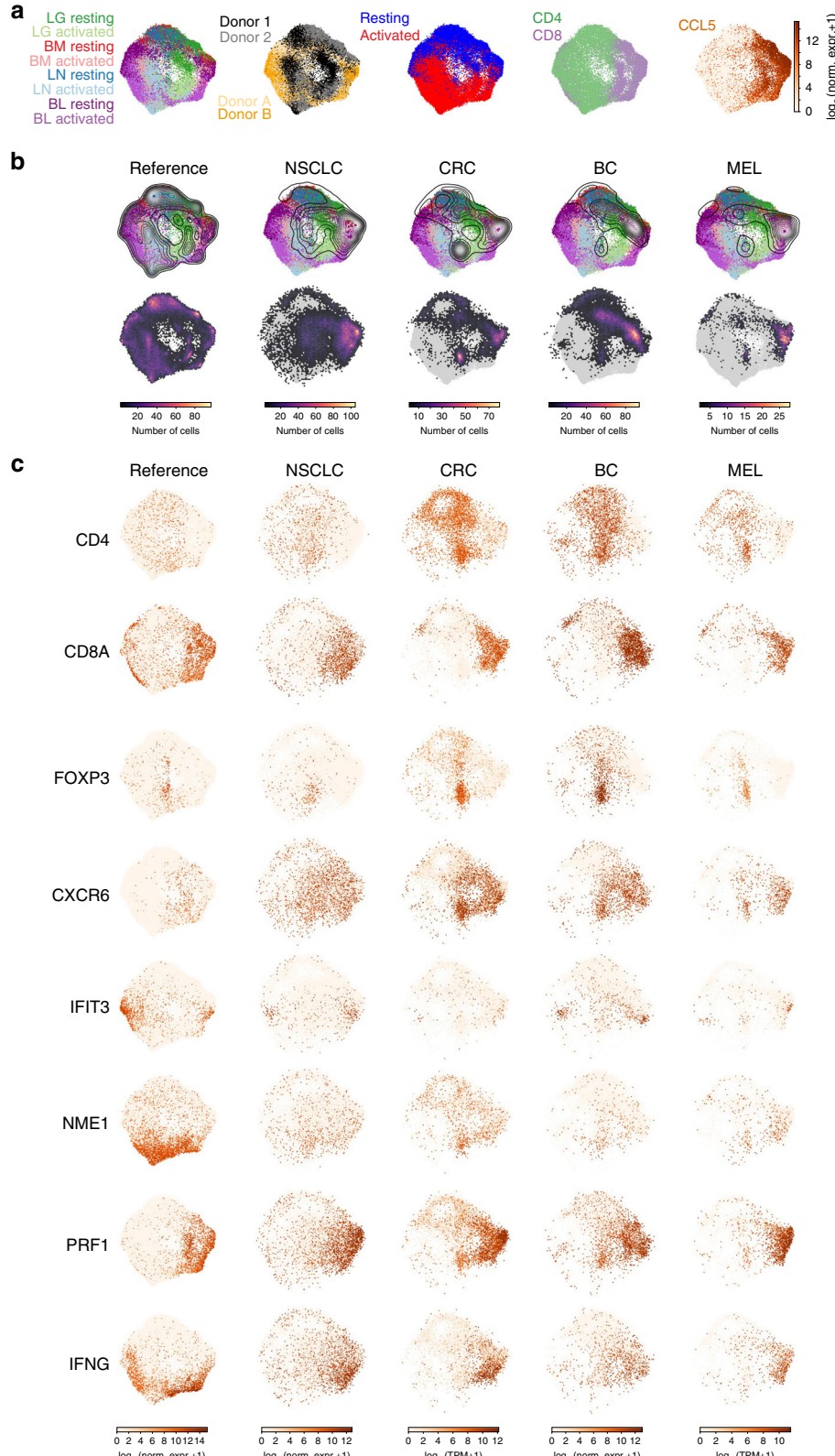

proliferation while CD8$^+$ T cells are defined by effector function. We further demonstrate that this high-resolution map of T cell homeostasis and activation across sites, lineages, and individuals can serve as a new baseline for defining human T cell states in disease.

The study of healthy human T cells has largely focused on blood, while the majority of T cells persist in diverse lymphoid, mucosal and barrier sites[8,61]. Human tissue T cells are largely memory subsets, comprising tissue-resident (TRM) and non-resident (TEM, TCM) populations; TRM predominate in mucosal sites, while TEM are found in spleen, LN and BM[13,33,62]. The transcriptional differences and functional relationship of these tissue-localized TEM to blood TEM has been unclear. Importantly, profiling using scRNAseq enabled unambiguous

**Fig. 6** Comparison of tumor-associated T cells to the reference map of healthy human T cell activation. **a** Merged UMAP embedding for the entire healthy T cell scRNAseq dataset in this study including resting and activated tissue T cells (two donors) and blood T cells (two individuals) colored by sample source, donor, resting/activated condition, CD4/CD8 status (CD4-enriched, green; CD8-enriched, purple), and CCL5 expression indicating TEM cells. **b** First row: merged UMAP embedding for the entire dataset overlaid with contour plots indicating kernel density estimates for the projection of T cells derived from organ/blood donors (column 1), non-small cell lung cancer (NSCLC) tissue (column 2), colorectal cancer (CRC) tissue (column 3), breast cancer (BC) tissue (column 4), and melanoma (MEL) tissue (column 5). Note that these probability densities can be compared within each projection, but cannot be quantitatively compared across projections. Second row: same as first row but overlaid with a two-dimensional hexbin histogram for each projection. Histograms have been normalized to account for differences in cell numbers across datasets and therefore can be compared quantitatively across projections. **c** Individual cells in the UMAP embedding (column 1) for the entire healthy T cell dataset and UMAP projections (columns 2–5) for NSCLC, CRC, BC, and MEL tissue T cells colored by expression of *CD4, CD8A, FOXP3* (Treg marker), *CXCR6* (TRM marker), *IFIT3* (IFN response marker), *NME1* (activation marker), *PRF1* (cytotoxic marker), and *IFNG*. Expression values are normalized for quantitative comparison within each dataset (i.e., column), but not across datasets

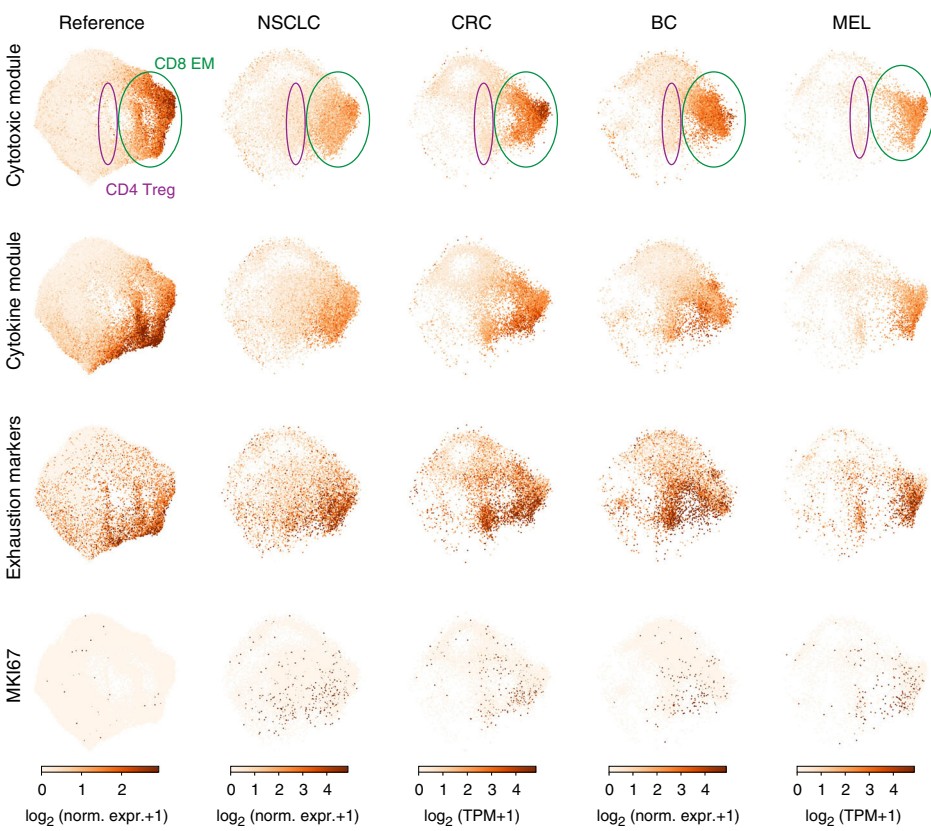

**Fig. 7** Expression of functional modules and exhaustion markers on control and tumor associated-T cells. Individual cells in the UMAP embedding (far left column) shown for the entire healthy T cell reference dataset and UMAP projections (remaining four columns) for NSCLC, CRC, BC, and MEL tissue T cells. UMAP projections are colored by the average expression of the top 70 genes in the Cytotoxic module, the top 70 genes in the Cytokine module, the average expression of a set of exhaustion markers (*PDCD1, CTLA4, LAYN, LAG3, TIM-3, CD244,* and *CD160*), and expression of the proliferation marker *MKI67*. Note that these expression values are normalized so that they can be quantitatively compared within each dataset (within each column), but not across datasets

assessment of T cell-intrinsic differences in tissue versus blood T cells. We show here that TEM from all tissue sites examined (LG, LN, BM) exhibit fundamental changes in expression of cytoskeletal, cell-matrix interaction, and proliferative genes compared to blood TEM cells, indicating alterations in cellular structure. These tissue-intrinsic expression patterns are in addition to TRM-associated genes identified in previous studies [13,33] or functional adaptations of T cells to specific tissue sites [12,63]. Whether T cells require these changes in gene expression to enter or reside within the tissue architecture, and if their loss of expression enables tissue T cell egress to circulation remains to be established.

Our results reveal conserved functional states for human blood and tissue-derived T cells. CD8[+] T cells segregate into two major effector subsets based on expression of genes involved in cellular cytotoxicity (Cytotoxic module) and myriad cytokines and chemokines (Cytokine module). These predominant effector states within activated human CD8[+] T cells are consistent with results showing that mouse CD8[+] T cell activation triggers an effector differentiation program[64,65]. We identified two major activation states that were not associated with effector function: one associated with proliferation and IL-2 production, and a second state enriched in CD4[+] T cells which is characterized by induction of multiple IFN-responsive genes and gene families including IFIT3,

MX1, IRF7, and others. Induction of this IFN-response state is due to TCR-mediated IFN-γ production (likely autocrine responses), and appears as a kinetic intermediate early after CD4[+] T cell activation, and prior to induction of the proliferative program. Identification of a functional state for T cells based on cytokine responses is distinct from T cell functional states that are typically defined based on cytokine secretion profile. We propose that the IFN-responsive state for human CD4[+] T cells may serve an autoregulatory function to temper high IFN levels produced by predominant memory responses, and ongoing responses to persistent viruses.

This scRNA-seq analysis provides a high-resolution map for human T cells from which to define T cell states in disease. We demonstrate this approach by projecting T cell profiles from human tumors onto our reference map. We identify predominant CD8[+] T cell effector populations, Tregs, and resting (but not activated) CD4[+] T cells in datasets derived from diverse tumor types (breast, lung, skin, colon). Interestingly, the tumor-associated CD8[+] T cells exhibited transcriptional features similar to healthy activated CD8[+] T cells including expression of multiple effector molecules such as perforin, IFN-γ and chemokines. We also examined the expression of multiple markers associated with exhaustion, a functionally hyporesponsive state found in tumor-infiltrating T cells targeted by checkpoint blockade immunotherapies[57,59,66]. Interestingly, exhaustion markers were upregulated along with CD8-associated cytokines in activated T cells from both healthy and tumor tissues, emphasizing the importance of obtaining baseline healthy profiles for high resolution analysis of T cells on the single cell level. Moreover, subsets of these CD8[+] T cells in all four tumors expressed higher levels of proliferation markers compared to healthy T cells, consistent with a recent report that T cells expressing exhaustion markers in melanoma exhibit aberrant proliferation[67]. This analysis can therefore enable precise identification of features of resting and activated T cells that are associated with tissues, activation and disease.

Our high-resolution analysis of human T cells across sites, lineages, and activation states provides insights into human T cell adaptations to tissues and their intrinsic activation properties. Limitations of the study include that the select tissues and donors profiled here may not include the full diversity of T cell transcriptional programs throughout the body, and that quantification of cell types may be subject to dissociation biases between the individual tissues[68,69]. Importantly, our dataset establishes a starting point for the integration of other T cell scRNA-seq datasets to ultimately capture the full breadth of T cells states in humans. International collaborative efforts like the Human Cell Atlas[70] are now underway, generating comprehensive scRNA-seq datasets profiling a diverse range of cells, including T cells and their transcriptional states. Recently developed computational tools including scVI[71], mutual nearest neighbors[72], Seurat v3[73], Conos[74], and Scanorama[75] will be useful for this integration and as a guide for future studies. In this way, our novel reference map can serve as a valuable resource for the ongoing study of human T cell immunity in disease, immunotherapies, vaccines and infections, with the ultimate goal of diagnosing, screening and monitoring immune responses.

## Methods

**Acquisition of human tissues and blood**. We obtained human tissues from deceased, brain-dead donors at the time of organ acquisition for clinical transplantation through an approved research protocol and MTA with LiveOnNY, the organ procurement organization for the New York metropolitan area. Obtaining tissue samples from deceased organ donors does not qualify as "human subjects" research, as confirmed by the Columbia University Institutional Review Board (IRB). Donors were free of chronic disease, cancer and chronic infections such as Hepatitis B, C, and HIV. Clinical and demographic data regarding organ donors

used in this study are summarized in Supplementary Table 1. We obtained peripheral blood from healthy consenting adult volunteers by venipuncture, through a protocol approved by the Columbia University IRB and have complied with all relevant ethical regulations for work with human participants.

**T cell isolation and stimulation**. Tissues acquired from donors were maintained in cold saline during transport to the laboratory, typically within 2–4 h of procurement. We isolated mononuclear cells from donor lungs, lung-draining lymph nodes (LN) and bone marrow (BM) as previously described[10,11]. Briefly, lungs were flushed with cold complete medium (RPMI 1640, 10% FBS, 100 U/ml penicillin, 100 μg/ml streptomycin, 2 mM L-glutamine) and left lateral basal segment of the lung was isolated. LN were isolated from the hilum, near the intersections of major bronchi and pulmonary veins and arteries, removing all fat. To obtain mononuclear cell suspensions, LN and lung tissues were mechanically processed using a gentleMACS tissue dissociator (Miltenyi Biotec), enzymatically digested (complete medium with 1 mg/ml collagenase D, 1 mg/ml trypsin inhibitor and 0.1 mg/ml DNase for 1 h at 37 °C in a mechanical shaker) and centrifuged on a density gradient using 30% Percoll Plus (GE Healthcare). BM was aspirated from the superior iliac crest. For BM and peripheral blood, we isolated mononuclear cells by density gradient centrifugation using Lymphocyte Separation Medium (Corning). T cells were enriched from all samples using magnetic negative selection for CD3[+] T cells (MojoSort Human CD3[+] T cell Isolation Kit; BioLegend), followed by a dead cell removal kit (Miltenyi Biotec), resulting in 80–90% purity. We cultured 0.5–1 × 10[6] CD3[+] enriched cells from each donor tissue for 16 h at 37 °C in complete medium, with or without TCR stimulation using Human CD3/CD28 T Cell Activator (STEMCELL Technologies). After stimulation, dead cells were removed as above before cell isolation for single-cell RNA-seq.

**Single-Cell RNA-seq**. Single-cell suspensions were loaded onto a Chromium Single Cell Chip (10x Genomics) according to the manufacturer's instructions for co-encapsulation with barcoded Gel Beads at a target capture rate of ~5000 individual cells per sample. We barcoded captured mRNA was barcoded during cDNA synthesis and converted the barcoded cDNA into pooled single-cell RNA-seq libraries for Illumina sequencing using the Chromium Single Cell 3′ Solution (10x Genomics) according to the manufacturer's instructions. All samples for a given donor were processed simultaneously with the Chromium Controller (10x Genomics) and the resulting libraries were prepared in parallel in a single batch. We pooled all of the libraries for a given donor, each of which was barcoded with a unique Illumina sample index, for sequencing in a single Illumina flow cell. All of the libraries were sequenced with an 8-base index read, a 26-base read 1 containing cell-identifying barcodes and unique molecular identifiers (UMIs), and a 98-base read 2 containing transcript sequences on an Illumina HiSeq 4000. Cell counts and transcript detection rates are summarized in Supplementary Table 2.

**Single-Cell RNA-seq data processing**. Prior to gene expression analysis, raw sequencing data was corrected for index swapping, a phenomenon that occurs during solid-phase clonal amplification on the Illumina HiSeq 4000 platform and results in cross-talk between sample index sequences. We corrected index swapping using the algorithm proposed by Griffiths et al[76]. First, we aligned the reads associated with each sample index to GRCh38 (GENCODE v.24) using STAR v.2.5.0 after trimming read 2 to remove 3′ poly(A) tails (>7 A's) and discarding fragments with fewer than 24 remaining nucleotides as described in Yuan et al[77]. For each read with a unique, strand-specific alignment to exonic sequence, we constructed an address comprised of the cell-identifying barcode, unique molecular identifier (UMI) barcode, and gene identifier. Next, we counted the number of reads associated with each address in each sample. Because of index swapping, we found that some addresses occurred in multiple samples at much higher frequencies than one would expect by chance. For the vast majority of addresses, there was a single sample containing most of the associated reads. If >80% of reads for a given address were associated with a single sample (e.g., a single index sequence), we kept all of the reads corresponding to that address in that sample and removed all of the reads associated with that address from all other samples[76]. We also identified addresses for which no sample contained >80% of the corresponding reads and removed all of these reads from all samples. After correcting for index swapping, we collapsed amplification duplicates using the UMIs and corrected errors in both the cell-identifying and UMI barcodes to generate a preliminary matrix of molecular counts for each cell as described previously[77].

We filtered the cell-identifying barcodes to avoid dead cells and other artifacts as described in Yuan et al[77]. Briefly, we removed all cell-identifying barcodes where >10% of molecules aligned to genes expressed from the mitochondrial genome or for which the ratio of molecules aligning to whole gene bodies (including introns) to molecules aligning exclusively to exons was >1.5. Finally, we also removed cell-identifying barcodes for which the average number of reads per molecule or average number of molecules per gene deviated by >2.5 standard deviations from the mean for a given sample.

**Computational identification of T cells**. Thoroughly removing non-T cells from the data set is complicated by technical issues such as molecular cross-talk, multiplet capture, and a broad coverage distribution. We developed a procedure to

remove non-T cells that accounts for these issues by identifying both individual cells and clusters of cells that are enriched in expression of a blacklisted gene set that is highly specific to contaminating cell types. We began by clustering the single-cell profiles within each sample using a pipeline that we reported previously[77,78]. Briefly, we identified highly variable genes that are likely markers of specific subpopulations by normalizing the molecular counts for each cell to sum to one, ordering all genes by their normalized expression values, and computing a drop-out score $ds_g$ for each gene $g$ defined as:

$$ds_g = \left| f_g - f_g^{max} \right| / \sqrt{f_g^{max}}, \tag{1}$$

where $f_g$ is the fraction of cells in which we detected $g$ and $f_g^{max}$ is the maximum $f_g$ in a 25-gene rolling window centered on $g$. We selected genes with $ds_g > 0.15$ or with $ds_g > 6\sigma_{ds} + <ds_g>$, where $\sigma_{ds}$ and $<ds_g>$ are the standard deviation and mean of the dropout score distribution. Using these genes, we computed a cell-by-cell Spearman's correlation, from which we constructed a k-nearest neighbor's graph ($k = 20$) and used this as input for the Phenograph[30] implementation of Louvain clustering to identify cellular subpopulations.

Next, we used the pooled normalization approach described by Lun et al. as implemented in the *scran* package by the *computeSumFactors* function to compute size factors for each cell[79,80]. We supplied the *computeSumFactors* function with the cluster identifiers obtained from Phenograph to account for cell type-specific coverage differences. Using the resulting normalized expression profiles, we identified Phenograph clusters with positive enrichment of average *CD3D* and *TRAC* expression and labeled these clusters as T cell clusters (Supplementary Fig. 1). Within each sample, we conducted differential expression analysis between all pairs of T cell and non-T cell clusters via the Wilcoxon rank-sum test using the SciPy function *ranksums* and Benjamini-Hochberg corrected *p*-values with the StatsModels function *multipletests* in Python, yielding *p* values $p_{adj}$. Finally, we established an initial blacklist of genes that are highly specific to the non-T cell clusters by taking any gene with $p_{adj} < 0.001$ and greater than 10 fold-enrichment in a non-T cell cluster for any of the above pairwise comparisons in any sample. To refine the blacklist and avoid including genes that are specific to T cell subsets found in only a limited set of samples or clusters, we also generated a whitelist of genes with positive enrichment in any T cell cluster. We removed any member of this whitelist from the initial blacklist to produce a final, refined blacklist containing 744 genes highly specific to contaminating cell types (Supplementary Data 1). As expected, genes on the final blacklist included markers of epithelial cells, dendritic cells, mast cells, B cells, neutrophils, and red blood cells.

The blacklist of genes was used to remove cells from the T cell clusters that are either improperly clustered (unlikely to be T cells) or potentially multiplets (a cell-identifying barcode co-encapsulated both T cells and non-T cells). Importantly, because of molecular cross-talk in scRNA-seq libraries from PCR recombination, we only considered a cell to be expressing a blacklisted gene if the average number of reads supporting the detected molecules was above a certain threshold. This threshold depends on the average depth to which we sequenced the libraries in a given sample. The distributions of the number of reads-per-molecule are generally bimodal for a given sample. We assume that the mode with lower read counts per molecule arises from PCR recombination in which a molecule originating from one cell receives the cell-identifying barcode of a different cell at an intermediate point in PCR, thereby resulting in a detected molecule supported by an unusually small number of reads (i.e., amplicons). We therefore considered the sample $h$ with the highest coverage (and therefore the clearest separation between the two modes) and took the minimum point between the two modes in the reads-per-molecule distribution to be the threshold number of reads per molecule, $T_h$, below which a detected molecule would be considered to arise from cross-talk. We extrapolated a reads-per-molecule threshold for each of the other samples $s$ as:

$$T_s = T_h * \left( \frac{RPM_s}{RPM_h} \right), \tag{2}$$

where $RPM_s$ is the average number of reads per molecule detected in sample $s$.

Finally, for each cell $c$ in a sample with threshold $T_s$, we computed $b_c$, the per-cell fraction of blacklisted genes detected with an average number of reads per molecule above $T_s$. As expected, $b_c$ was typically bimodally distributed within each sample (Supplementary Fig. 1e). The vast majority of cells in the lower mode were in the T cell clusters described above, while the high mode was composed mainly, but not exclusively, of cells from non-T cell clusters (Supplementary Fig. 1e). In each sample, we fit a Gaussian to $b_c$'s distribution across cells assigned to T cell clusters and established a threshold at two standard deviations above the fitted mean. We considered any cell with $b_c$ above this threshold and any cell that clustered among the non-T cell clusters to be a non-T cell and discarded these cells from all downstream analysis.

**Course-grained clustering of T cells from each donor.** Once we had identified the T cells from each sample using the methodology described above, we merged resting and activated samples from all of the tissues in each donor and clustered the T cells from the two donors separately to generate Fig. 1b, c. We used the methodology described above to identify a set of highly variable genes for each sample (including the blood samples), and then merged those sets to generate a large list of 315 highly variable genes (Supplementary Data 2) with which we clustered the merged samples from both donors. We computed Louvain clusters from the two

merged data sets with $k = 12$ and a minimum cluster size of 100 cells using a k-nearest neighbors graph constructed from the Spearman's correlation matrix calculated using the 315 highly variable genes. We used the Python implementation Uniform Manifold Approximation and Projection (UMAP)[31] to produce the two-dimensional projections shown in Fig. 1b, c. To obtain *CD4/CD8* ratios for each cluster, we first computed the expression level of *CD4* and *CD8A* in each cell using the normalized counts from *computeSumFactors* as described above. For both *CD4* and *CD8A*, we then computed the average log2(normalized counts + 1) for each cluster and normalized this value by the average log2(normalized counts + 1) for all cells. We then took the log-ratio of these values for *CD4* and *CD8A* to generate Fig. 1b, where all the cells in each cluster are labeled with the cluster's log-ratio. Differentially expressed genes for cells in each cluster versus all other cells were determined using a binomial test[81] (Supplementary Data 3, 4).

**Blood projection analysis.** To project the data obtained from blood T cells onto the tissue-derived profiles from each organ donor, we first merged the scRNA-seq profiles from both blood donors. We note that the scRNA-seq data from blood were subjected to the same computational procedure described above for eliminating non-T cell profiles. We used the same highly variable gene set (Supplementary Data 2) that was used in the original UMAP model of each organ donor to compute a Spearman's correlation matrix between the blood and tissue profiles. We then projected the blood T cell profiles onto the UMAP embeddings for each of the two organ donors using the *transform* function in UMAP. We note that the organ donor UMAP embeddings used for this analysis are slightly different from what appears in Fig. 1b, c, because a small number of genes in the highly variable gene set were eliminated due to lack of expression in the blood. We also note that a small modification to the UMAP source code was needed to accommodate the use of Spearman's correlation as a similarity metric (available at https://github.com/simslab/umap_projection).

We confirmed our findings using scmap[34], a previously published scRNA-seq data projection algorithm. When projecting blood T cells onto the tissue T cells from Tissue Donors 1 and 2, scmap yielded projections that were consistent with UMAP (Supplementary Fig. 4a, b, e, f) with projection coordinates that were highly correlated across both data sets (Supplementary Fig. 4c, d, g, h).

To generate the cell number heatmaps in Fig. 2 and Supplementary Fig. 3, we first computed a centroid position in the UMAP embedding for each condition, subset and tissue combination in the tissue data based on the Louvain clustering described above for Fig. 1b, c. For example, for Donor 2, we computed the average position of LG-, BM-, and LN-derived resting. We then identified the nearest condition, subset and tissue combination for each cell in the blood samples based on the Euclidean distance between a given blood-derived cell's position in the UMAP model (following projection of the blood data onto the tissue UMAP model) and each centroid position. The heatmaps summarize the results of these calculations, providing the number of blood-derived cells that are closest to each condition, subset and tissue combination in the organ donor data.

**Comparison of TEM cells from tissue and blood.** To identify a tissue-specific T cell signature, we compared the expression profiles of effector memory cells from resting LG, BM, and LN T cells from the two tissue donors to resting blood T cells from the two blood donors. We found *CCL5* to be an extremely highly expressed marker of effector-memory (TEM) cells that exhibited strong anti-correlation with *SELL*, a marker of non-effector memory cells, in all of our resting samples (Supplementary Fig. 5a). We also found that the average number of reads per molecule for *CCL5* was bimodally distributed, consistent with spurious detection of *CCL5* in a population of cells due to PCR recombination (Supplementary Fig. 5b). For each sample, we used the point between these two modes where the probability density was minimal as a threshold for the minimum average number of reads per molecule of *CCL5* required for a cell to be considered positive for *CCL5*. For each sample, we normalized the matrix of molecular counts for the *CCL5*+ TEM cells using the *computeSumFactors* function in *scran* to compute size factors for each cell[79,80]. For each tissue site, we then identified differentially expressed genes for all four pairwise comparisons of resting tissue to resting blood *CCL5*+ T cells (tissue donor 1 vs. blood donor A, tissue donor 2 vs. blood donor A, etc.) using the Wilcoxon rank-sum test with the SciPy function *ranksums* and computed Benjamini-Hochberg corrected *p*-values with the StatsModels function *multipletests* in Python after removing genes from the blacklist described above (Supplementary Data 1). For each tissue, we took all genes with $p_{adj} < 0.05$ and fold-change > 2 in all 4 pairwise comparisons to comprise a tissue-specific effector memory T cell signature (Fig. 3).

Next, all of the genes in the tissue-specific effector memory T cell signature and computed the average normalized expression of the resulting gene set to obtain Fig. 3d. Z-scored normalized expression for each of these genes appears in the heatmap in Fig. 3e for each site/donor, which also includes a set of blood T cells with outlier expression of the tissue-enriched gene signature (blood T cells with average expression within one standard deviation of that of the tissue T cells as indicated by the dashed line in Fig. 3d). For Fig. 3f, we took all genes with $p_{adj} < 0.05$ (Wilcoxon and correction described above) and two-fold higher expression in either the tissue- or blood-associated T cells in all four pairwise comparisons. We then constructed a Spearman's correlation coefficient between pairs of resting

CCL5$^+$ cells in the dataset across these genes and used this to generate a UMAP embedding.

**Analysis of tissue T cell signatures in other datasets**. We used Gene Set Enrichment Analysis (GSEA) to assess the enrichment of the tissue-associated T cell signature in T cells profiled from the bone marrow of living donors and its depletion in T cells profiled from two additional blood samples in Supplementary Fig. 6. We obtained 10x Genomics Chromium scRNA-seq profiles of bone marrow from 20 individuals from GEO accession GSE120221[37]. We merged the data from all of the samples and clustered the merged scRNA-seq profiles using Phenograph as described above. To computationally isolate TEM cells, we took all CCL5$^+$ cells that occurred in any Phenograph cluster that was positively enriched in TRAC expression. We then compared the expression profiles of CCL5$^+$ T cells from the bone marrow of living donors to the resting, CCL5$^+$ T cells from the two blood donors from this study (Donors A and B) using the Wilcoxon rank-sum test as described above. After ranking all genes for which a test statistic could be computed by fold-change (comparing bone marrow to blood), we used GSEA (pre-ranked, "classic" mode with 10,000 permutations) to calculate the enrichment of the tissue-associated T cells signature among the differentially expressed genes. We used the Java implementation of GSEA that is freely available from http://software.broadinstitute.org/gsea/index.jsp.

To assess the depletion of the tissue-associated T cell signature in other blood data sets relative to the tissue samples collected here, we obtained 10x Genomic Chromium scRNA-seq data from PBMCs of a healthy donor and purified T cells of a healthy donor from 10x Genomics (Donor C PBMCs: pbmc8k data set from https://support.10xgenomics.com/single-cell-gene-expression/datasets/2.1.0/pbmc8k; Donor D purified T cells: t_4k data set from https://support.10xgenomics.com/single-cell-gene-expression/datasets/2.1.0/t_4k). We performed clustering separately on the two samples using Phenograph and computationally isolated TEM cells as described above for the live donor bone marrow samples. We then performed differential expression analysis using the Wilcoxon rank-sum test to compare TEMs from Donors C and D to those in each tissue from the resting samples collected for this study. For each tissue site, we merged the resting TEMs from Donors 1 and 2. Finally, we performed GSEA as described above to assess the depletion of the tissue-associated T cell signature in Donors C and D for each tissue site comparison.

**Single-cell hierarchical poisson factorization analysis**. We applied Single-cell Hierarchical Poisson Factorization (scHPF), a method that we recently reported for de novo discovery of gene expression signatures in scRNA-seq data, to the merged activated and resting cells for each tissue and donor[38]. Given a molecular count matrix, scHPF identifies a small number of latent factors that explain both continuous and discrete expression patterns across cells. Each gene has a score for each factor, quantifying the gene's contribution to the associated expression pattern. Likewise, each cell assigns a score to each factor, which reflects the contribution of the factor to the observed expression in the cell.

We applied scHPF to each tissue and blood sample after merging their respective resting and activated datasets. We considered only genes with GENCODE protein coding, T cell receptor constant or immunoglobulin constant biotypes, excluded genes on the previously described blacklist, and removed genes detected in fewer than 0.1% of cells in a given merged dataset. scHPF (version 0.1) was run with default parameters for seven values of $K$, the number of factors, equal to all values between 6–12, inclusively. This resulted in seven candidate scHPF factorizations per merged dataset. We then selected $K$ to avoid factors with significant overlap in their gene signatures. For each dataset and value of $K$, we calculated $N_K$: the maximum pairwise overlap of the 300 highest-scoring genes in each factor for the corresponding scHPF model. We considered overlap significant if $p < 0.05$ by a hypergeometric test with a population size equal to the number of unfiltered genes in the tissue sample and $N_K$ observed successes. Finally, for each dataset, we selected the model with maximum $K$ such that $p >= 0.05$ (Supplementary Fig. 7). This procedure resulted in eight factorizations: six from tissue donors (lung, BM, and LN from each of two organ donors) and two factorizations from the blood of living donors. We defined each factors' CD4/CD8 bias as the log2 ratio of its mean cell score in CD4$^+$ and CD8$^+$ T cells.

To discover common patterns of expression across tissues and donors, we performed unsupervised clustering of all factors for tissue-derived cells. First, we calculated Pearson correlation on the union of the fifty highest and lowest scoring genes in each factor for each tissue factorization (2291 genes total) using the Python pandas package's *DataFrame.corr* function. Next, we hierarchically clustered the factor-factor correlation matrix using *scipy.cluster.hierarchy.linkage* with method = 'average' and *scipy.cluster.hierarchy.dendrogram* (Supplementary Fig. 8a). This defined clusters of tightly correlated expression patterns, which we call expression modules. We focused on seven modules (out of nine) whose factors had mean pairwise correlations greater than 0.25. Most modules contained at least one factor from each tissue and donor. To identify the top genes in each module (Fig. 4a, Supplementary Data 5), we ranked genes by their mean gene score across all constituent factors. The CD4 IFN response module contained two factors from Donor 2 BM; however, one of the two factors was far more tightly correlated with the rest of the factors in the module than the other. As the top genes in the module

were nearly identical with and without the less tightly-correlated factor, we excluded it from the module in downstream analyses for clarity.

**Activation trajectory analysis**. We used the factorizations described above to compute T cell activation trajectories by diffusion component analysis. We first converted the cell score matrix obtained from the factorization of each resting/activated merged tissue or blood sample into a cell-by-cell Euclidean distance matrix. We then extracted the distance submatrices corresponding to the CD4 and CD8 clusters in each sample as defined from the merged analysis of all samples from each donor described above. We used the two resulting distance submatrices to compute diffusion components for CD4 and CD8 activation with the C++ Accelerated Python Diffusion Maps Library (DMAPS) with a kernel bandwidth of four. The diffusion maps shown in Fig. 4b–e each show the first two diffusion components which we define as the two diffusion eigenvectors with the second- and third-highest eigenvalues scaled by the diffusion eigenvector with the largest eigenvalue.

**Flow cytometry and proliferation assays**. To evaluate the expression of T cell surface markers by flow cytometry, we incubated tissue and blood cell suspensions with Human TruStain FcX (BioLegend) and stained with following fluorochrome-conjugated antibodies: CD3 (UCHT1, BD Biosciences; OKT3, BioLegend), CD4 (SK3, BD Biosciences; SK3, Tonbo Biosciences), CD8 (SK1, BioLegend; RPA-T8, BD Biosciences), CCR7 (G043H7; BioLegend), CD45RA (HI100; BioLegend), CD25 (BC96; BioLegend), CD127 (A019D5; BioLegend), CD69 (FN50; BioLegend), CD103 (Ber-ACT8; BioLegend), CD45 (HI30; BioLegend), and Fixable Viability Dye eFluor 780 (eBioscience). For stimulation/proliferation assays, we magnetically enriched for CD3$^+$ T cells from single cell suspensions, stained cells with Cell Proliferation Dye eFluor 450 (eBioscience), and cultured cells for up to 120 h with or without TCR stimulation as above. At indicated time points, we performed intercellular staining of NME1 (11615-H07E; Sino Biological) using a Foxp3/Transcription Factor Staining Buffer Kit (Tonbo Biosciences) for fixation and permeabilization of cells according to manufacturer's instructions. We acquired cell fluorescence data using a BD LSR II flow cytometer and used FCS Express (De Novo Software) for analysis. The results are summarized in Supplementary Fig. 2 and the gating strategy is shown in Supplementary Fig. 18a.

**Quantitative real-time PCR**. PBMC were magnetically enriched for CD3$^+$ T cells, and sorted for live CD4$^+$ and CD8$^+$ T cells (singlets, FSC$^{low}$SSC$^{low}$, and Viability Dye$^-$) using a BD Influx cell sorter (Supplementary Fig. 18b). Sorted cells were cultured in complete medium with or without anti-CD3/anti-CD28 stimulation as above for 2–72 h. For dissecting the contribution of type I and type II IFN signaling to gene expression, cells were pre-incubated with Human Type 1 IFN Neutralizing Antibody Mixture (PBL Assay Science, Cat# 39000-1) according to manufacturer's instructions, or 1 µg/mL of both anti-IFNγ (R&D Systems, MAB285, clone # 25718) and anti-IFNγR1 (R&D Systems, MAB6731, clone # 92101). As a control, CD4$^+$ T cells were activated with 1000 units/mL of recombinant human IFNα2 (PBL Assay Science, Cat#11101-1) or 10 ng/mL recombinant human IFNγ (Peprotech, Cat# 300-02). Control and stimulated CD4$^+$ and CD8$^+$ T cells were harvested at indicated time points and RNA isolated using a RNeasy Micro Kit (Qiagen) with on-column DNase digestion. We converted RNA to cDNA via SuperScript IV VILO Master Mix (Invitrogen) and performed quantitative real-time PCR (qPCR) on a Viia 7 Real-Time PCR system (Applied Biosystems) using TaqMan Gene Expression Assays (NME1 Hs00264824_m1; IL2RA Hs00907777_m1; IFIT3 Hs00155468_m1; TBP Hs00427620_m1) and TaqMan Fast Advanced Master Mix, all from ThermoFisher Scientific. Quntitative PCR (qPCR) reactions were set up according to manufacturer's instructions and fold changes between stimulated and unstimulated cells at each time point were calculated using the ΔΔ cycle threshold method in ExpressionSuite Software (ThermoFisher Scientific) with TBP as a reference gene.

**Tumor-associated T cell projection analysis**. We projected scRNA-seq profiles of tumor-associated T cells from four different tumor types onto a UMAP embedding of resting and activated T cells from our combined tissue and blood data set using the methods described above for projecting blood T cells onto embeddings of the tissues. Briefly, we used the highly variable gene set from Supplementary Data 2 to generate a UMAP embedding of our tissue/blood data from a Spearman's correlation matrix. We did not find any qualitative differences between UMAP embeddings when donor-specific genes were removed (Supplementary Fig. 19). We then projected the tumor-associated T cell profiles onto this embedding using the *transform* function in UMAP. Tumor-associated T cells from non-small cell lung cancer (NSCLC)[53] and breast cancer (BC)[52], which were profiled using the 10x Genomics Chromium platform, were obtained from https://gbiomed.kuleuven.be/scRNAseq-NSCLC and GEO accession GSE114724 (samples BC09, BC10, and BC11, respectively). For these two data sets, we used the UMI-corrected molecular counts provided by the authors. T cells from colorectal cancer (CRC)[51] and melanoma (MEL)[27], which were profiled using SMART-seq, were obtained from GEO accessions GSE108989 and GSE120575 (pre-treated samples only). For these two data sets, we used the TPM values provided by the authors. We note that the tissue/blood embedding was re-computed for each projection and is

therefore slightly different in each case because not all of the processed data sets from the tumor studies contained all of the genes in Supplementary Data 2.

The resulting projections are displayed in Fig. 6 in three different ways. In the top row, the projections are displayed as contour plots of estimated probability density (kernel density estimates) with a maximum of 14 contours. In the second row, we used a hexbin two-dimensional histogram of the number of cells in each bin with the colorbars normalized such that the intensity can be compared across samples (e.g., scaled so that the melanoma projection can be compared to the CRC projection). Finally, we also show where individual tumor-associated T cells project in subsequent rows along with gene expression values for several key markers. In Fig. 7, we show the average expression of several canonical exhaustion markers in individual cells. The markers used for this analysis were *PDCD1*, *CTLA4*, *LAG3*, *LAYN*, *TIM-3*, *CD244*, and *CD160*. We applied the same methodology to project the tumor-associated T cell profiles onto independent UMAP embeddings for each donor as shown in Supplementary Figs. 11–14.

As above in Fig. 2, we validated the tumor-associated T cell projection analysis using scmap[34]. Mapping the tumor-associated T cells onto our reference tissue and blood T cell dataset using *scmap* generated projections that were consistent with UMAP, with coordinates that were highly correlated (Supplementary Fig. 15).

**Analysis of T cells from dengue virus-infected patients**. To analyze the expression of *IFIT3* and *NME1* in the context of virus infection, we analyzed scRNA-seq profiles of peripheral blood from dengue virus-infected patients (GSE116672). We clustered the data using the methodology described above and isolated T cell clusters based on enrichment of *TRAC* expression. We then generated the UMAP embedding shown in Supplementary Fig. 10 using the methodology described above and the same 315-gene set used throughout this study (Supplementary Data 2).

**Reporting summary**. Further information on research design is available in the Nature Research Reporting Summary linked to this article.

## Data availability

All scRNA-seq data are available on the Gene Expression omnibus (GEO) under accession number GSE126030 [https://www.ncbi.nlm.nih.gov/geo/query/acc.cgi?acc=GSE126030]. We have included a pre-processed and filterable data table containing a matrix of molecular counts for all cells profiled in our study. Cell-identifying barcodes, UMAP coordinates and other characteristics (tissue origin, stimulation condition, CD4 or CD8 status and CCL5 expression) for cells designated as T cells (as described in Methods) are included in the Source Data file for Fig. 6 of this study. The source data underlying Figs. 1c, 3a–e, 4a, 5a, c, d, e and 6 are provided in the Source Data file.

## Code availability

The computer code for marker selection, clustering, and differential expression is available at https://github.com/simslab/cluster_diffex2018; the code for scHPF is available at www.github.com/simslab/scHPF. The code for umap projection analysis is available at: https://github.com/simslab/umap_projection.

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

## Acknowledgements

This work was supported by the US National Institutes of Health (NIH) (Grant nos. AI128949, AI106697 to D.L.F.; AI128949, AI106697 and Chan Zuckerberg Initiative Pilot Projects for the Human Cell Atlas to P. A. Sims.). P.A.Sz. was supported by the American Association of Immunologists (AAI) Intersect Fellowship Program for Computational Scientists and Immunologists. These studies were performed in the Columbia Center for Translational Immunology (CCTI) Flow Cytometry Core funded in part through an S10 Shared Instrumentation Grant from the NIH (S10RR027050), with the excellent technical assistance of S.-H. Ho. We thank the Columbia Single Cell Analysis Core for their assistance with scRNA-seq library preparation and data analyses. We gratefully acknowledge the generosity of the organ donor families and Dr. Amy Friedman and the LiveOnNY transplant coordinators and staff for making this study possible.

## Author contributions

P.A.Sz. designed, executed and analyzed experiments; H.M.L. developed the scHPF module analysis approach, H.M.L. and P.A.Si. performed computational analysis; T.E.S. obtained tissues from donors; M.M., M.E.S., processed tissues and optimized protocols, E.C.B. constructed and sequenced the scRNA-seq libraries, J.Y. and Y.L.C. optimized scRNA-seq experiments, P.D. and P.T. provided technical assistance; P.A.Sz., H.M.L., D.L.F., and P.A.Si. analyzed data, wrote and edited the manuscript; D.L.F. and P.A.Si. designed and coordinated the study.

## Competing interests

The authors declare no competing interests.
