## [Peer Review File · Nature Communications]

Reviewers' comments:

Reviewer #3 (Remarks to the Author):

revised ms is acceptable

Reviewer #4 (Remarks to the Author):

Szabo and colleagues generated a scRNA-seq dataset comprising more than 50,000 resting and activated T cells from lung, lymph nodes, and bone marrow samples collected from two deceased donors, and from blood samples collected from two healthy donors. Through bioinformatics analysis, they identified:

- A tissue-specific gene signature that distinguishes tissue-resident T cells from circulating T cells;
- T-cell transcriptional fingerprints shared by blood- and tissue-specific T cells;
- Two novel candidate markers of CD4+ T cell activation that they could validate in vitro.

They further showed how to leverage the data generated in this study (referred to as "reference map") to investigate the cellular composition of external scRNA-seq datasets.

The amount of experimental and bioinformatic work is impressive and sheds some light on an important and debated topic in immunology: T-cell phenotypes and transcriptional programs in different tissue contexts.

However, I have a few major concerns regarding:

- The possibility to consider this dataset as a definitive T-cell reference map in blood and tissues;
- The robustness of the bioinformatics methods and results given the state of the art in scRNA-seq data analysis;
- The clarity of the described analytical approaches.

In the following, I first motivate and detail these concerns, and then provide some suggestions on other major and minor points.

REFERENCE MAP

The authors stress in several sections of the manuscript how this dataset can be considered a reference map for future studies. However, due to the limitations in terms of sample size (two donors per tissues) and total tissues (four), these data might under-represent the real biological variability of T-cell transcriptional programs in different individuals and tissues. Thus, I would suggest toning down this kind of claims (e.g. "comprehensive baseline of healthy T cells states in humans", l. 76-77) and discuss openly possible limitations of the present study in the Discussion section.

Despite these limitations, this dataset represents a valuable resource that should be rendered more easily usable, especially if the intent is to propose it as a reference for future single-cell studies. For instance, how can this "reference map" be reconstructed easily from the data available on GEO and used for interpretation of the external dataset? This kind analysis could be made easily accessible by providing the preprocessed reference in a convenient format, together with some (R?) code to run the comparison (e.g. to generate plots similar to those of Fig. 6b).

Alternatively, tools based on reference count matrices, like scmap (PMID: 29608555), could be used. I would include some information in this sense to guide the readers in this kind of analysis.

METHOD ROBUSTNESS

The authors present the bioinformatics methods adopted as “innovative” (l. 87), which I agree with. However, the field of scRNA-seq data analysis is fast-paced and far from being standardized. Many methods are available for the same analytical task, rely heavily on parameter setting and data preprocessing, and often return different results. Thus, the robustness of methods and results has to be endorsed by using orthogonal methods.

For instance, I would suggest considering these methods in parallel with the analyses already performed:

- Projection of a query scRNA-seq dataset on a reference dataset with scmap (PMID: 29608555);
- Integration of different datasets with Seurat 3 (PMID: 31178118, see the description about the identification of “anchor” cells);
- Single-cell classification with SingleR (PMID: 30643263).

These methods are also useful to confirm the projections of Fig. 2 and 6.

Finally, approaches like those implemented Seurat might be useful to integrate and harmonize several T-cell datasets to really provide an ultimate reference map. This is probably out of the scope of the current manuscript, but still worth mentioning in the Discussion section as a future development.

CLARITY OF METHOD DESCRIPTION

In the Methods section, the description of the cell-type identification and scHPF analyses should be improved to facilitate understanding and reproducibility.

I understand that cell-type identification was performed in three major steps:

1. Clustering for each donor separately, all tissues together, expecting $K=12$ clusters.
2. Cell-type classification.
3. Further selection of cell-type specific genes to be shown in Fig. 1c.

The Methods section should describe clearly how steps 2 and 3 were performed, so to allow reproducibility.

Regarding step 2, I point out that recent best practices (PMID: 31217225) suggest integrating automatic (like SingleR) and manual approaches (based on prior knowledge of marker genes) to annotate cell types in large datasets like this one.

Similarly, for scHPF analysis:

1. Identification of factors specific for each sample (i.e. tissue-donor) using scHPF.
2. Clustering of the factor-factor correlation matrix to identify major modules.
3. Annotation/interpretation of modules.

Regarding step 2, I think the terminology is a bit confusing: are “modules” and “models” the same thing? Also, the sentence about the seven models mentioned at l. 629-630 is probably too premature and confusing because, at this point, only factors are identified, not modules.

Step 3 requires a careful description of how the modules were annotated and why two modules could not be annotated. If possible, module-specific genes should be reported also for these two modules in Suppl. Table 7.

Please, also double check that the gene sets identified and used in Fig. 1c (differentiating T cell lineages), Fig. 3a-c (DEGs), and Fig. 3e (tissue signature) are fully reported in the Supplementary Material and referenced in the main text, and that the exact procedures for their identification is described in a Methods sub-section. The underlying methods and final gene-set results are not straightforward to picture out from the current manuscript.

MAJOR POINTS

I do not agree with the following sentence (l. 113-115): "T cells from BM and LN co-clustered while LG T cells were more distinct (Fig. 1b), consistent with T cell subset composition differences in these sites from phenotype analysis".

UMAP plots show clustering of cells based on similarities of the transcriptional profiles of single-cells, whereas sample composition is an overall quantification of the different cell subtypes. These two aspects do not have a 1:1 correspondence but rather provide complementary insights. Also, the cellular composition is not easy to visualize from overlapping UMAP points.

I would suggest rephrasing, as well as integrating both types of information in these plots (e.g. Fig. 6b), by showing both the UMAP plot and, next to it, a small, stacked barplot representing the cellular composition.

In addition, it should be mentioned in the text that the quantification of cell subtypes from scRNA-seq data might be biased by single-cell dissociation efficiencies (PMID: 30345255).

Being this work mainly based on dry- rather than wet-lab work, I am wondering whether you can confirm the results of your "kinetic analysis" of CD4+ T cells (p. 12) also computationally, for instance through analysis of CD4+ T cell "velocity" (PMID: 30089906) and evolution in pseudotime of the expression of the IFIT3 and NME1 genes (PMID: 24658644).

The transcriptional profiles of tumor-infiltrating CD8+ T cell were found to be consistent across all tissues, whereas the CD4+ T cell phenotypes were more variable. Could this result be due to a bias in the set of highly variable genes considered, as many of them were CD4-T-cell-specific, and much fewer CD8-T-cell-specific (as shown in Fig. 1c heatmap)?

Some interesting and original results might be put in a better light by discussing them more deeply. For instance, is there anything in the literature supporting the association found in this study between localization of T cells and structural changes?

Also, Treg transcriptional fingerprints have been previously reported to be very plastic and variable across tissues, and in your dataset seem to also differ between donors (Fig. 1b-c, resting in D1, activated in D2). Some discussion about the Treg phenotypes found in the different tissues of the present study would be extremely interesting for the scientific community.

Besides Tregs, there is a lot of debate about the variability of immune-cell transcriptional profiles in different tissue and disease contexts, and on whether these differences should be taken into account by deconvolution methods for the quantification of cell subtypes from bulk transcriptomic data (PMID: 29541787). As you identified both tissue-dependent and -independent markers, a discussion of these aspects might provide valuable guidance for the researchers in this field.

I agree that the current study can provide a basis to study T-cell phenotypes in cancer, but I find the analysis proposed too limited and not really conclusive. Thus, I would suggest investigating a bit more its relevance for cancer immunology.

One idea could be, for example, to test the prognostic and predictive value of the selected CD4-T-cell-activation markers using data from TCGA or from patients treated with immune checkpoint blockers, respectively.

MINOR POINTS:

"CD3+T" (and similar ones) should be corrected to "CD3+ T" (with spaces after "+").

Please provide some rationale for the parameter setting presented in the Methods section (e.g. Louvain clustering with K=12 or DMAPS kernel bandwidth of 4).

I would put the heatmaps of Fig. 1c in a separated panel (Fig. 1d) and use the same ordering of genes for the two heatmaps so to ease comparison.

In the second plot of Fig. 3f, the palette does not allow distinguishing well the groups.

For the reasons explained above, I would not call the tissue gene signature "universal" (l. 148).

Fig. 3a-c and the relative captions are a bit confusing. How were the red genes selected? The "several highlighted in red" explanation does not help in this sense. I would rather show the differentially expressed genes selected, together with the lines identifying the thresholds specified in the methods: $p\text{-adj} < 0.05$, $\log\text{FC} > 2$ (l. 573).

What is exactly the Source Data (l. 961-962)? What do the clusters (1-10) on the columns of this file correspond to?

Please, revise the following sentence (l.953-956) to better clarify what is presented in Fig. 3d and what is the meaning of "certain genes": "Heatmap shows z-scored average expression for all genes in the tissue signature from (d) among the resting CCL5+ 954 T cells from each tissue and blood sample plus that of the rare blood subpopulation from (d), which expresses high levels of certain genes".

I would add x- and y-axis labels to Suppl. Fig. 7a.

Where is it explained in the Methods how were the diffusion maps built?

I would add the number of samples/replicates to Fig. 5 panels.

What is the intended meaning of "biased" in the sentence at l. 261?

How were exhaustion markers quantified as a single variable in Fig. 7?

The association between a subset of CD8+ T cells with proliferation markers discussed at l. 312-314 would be probably more easily measured and visualized by using the "proliferation module" previously identified, instead of the single markers (which might also be sensitive to dropout).

What is the intended meaning of "relationship" at l. 333? Are you referring to transcriptional/functional differences between the two types of TEM? Please, clarify.

Throughout the manuscript, I would suggest referring to "transcriptional profiles" rather than "functional states" of T cells, as the actual functional states were not assessed in the study, but only the transcriptomes that could underlie them.

What is the rationale behind the selection of models with $p\text{adj} \geq 0.05$ (l. 636)?

Point by Point Response to Reviewers

NCOMMS-19-16425A

“A single-cell reference map for human blood and tissue T cell activation reveals functional states in health and disease”

Corresponding Authors: Donna L. Farber and Peter A. Sims

We are submitting a revised version of our manuscript for consideration that addresses Reviewer 4’s comments. We have included new data and analysis, along with corresponding revisions and clarifications in the revised manuscript, indicated by underlining in the text. Below, we present an itemized response to each comment, including a description of corresponding revisions to the text and figures.

Reviewer 4:

Reviewer 4 enumerated the main findings of our study, stating that “The amount of experimental and bioinformatic work is impressive and sheds some light on an important and debated topic in immunology: T-cell phenotypes and transcriptional programs in different tissue contexts. However, I have a few major concerns regarding: The possibility to consider this dataset as a definitive T-cell reference map in blood and tissues; The robustness of the bioinformatics methods and results given the state of the art in scRNA-seq data analysis; The clarity of the described analytical approaches.

Response: We appreciate the positive comments of the reviewer regarding the key novel findings and the importance of the study. We have separately addressed the major concerns as itemized below.

REFERENCE MAP

1) The authors stress in several sections of the manuscript how this dataset can be considered a reference map for future studies. However, due to the limitations in terms of sample size (two donors per tissues) and total tissues (four), these data might under-represent the real biological variability of T-cell transcriptional programs in different individuals and tissues. Thus, I would suggest toning down this kind of claims (e.g. “comprehensive baseline of healthy T cells states in humans”, l. 76-77) and discuss openly possible limitations of the present study in the Discussion section.

Response: We have removed the word “comprehensive” and qualified the text as suggested (pgs. 4, 16), and added a brief discussion about the limitations of our study and the potential of our reference map to inform studies on human T cells in health and disease (pg. 18 and below).

“Limitations of the study include that the select tissues and donors profiled here may not include the full diversity of T cell transcriptional programs throughout the body, and that quantification of cell types may be subject to dissociation biases between the individual tissues^{1,2}. Importantly, our dataset establishes a starting point for the integration of other T cell scRNA-seq datasets to ultimately capture the full breadth of T cells states in humans. International collaborative efforts like the Human Cell Atlas³ are now underway, generating comprehensive scRNA-seq datasets profiling a diverse range of cells, including T cells and their transcriptional states. Recently developed computational tools

including scVI⁴, mutual nearest neighbors⁵, Seurat v3⁶, Conos⁷, and Scanorama⁸ will be useful for this integration and as a guide for future studies. In this way, our novel reference map can serve as a valuable resource for the ongoing study of human T cell immunity in disease, immunotherapies, vaccines and infections, with the ultimate goal of diagnosing, screening and monitoring immune responses.”

2) *Despite these limitations, this dataset represents a valuable resource that should be rendered more easily usable, especially if the intent is to propose it as a reference for future single-cell studies. For instance, how can this “reference map” be reconstructed easily from the data available on GEO and used for interpretation of the external dataset? This kind analysis could be made easily accessible by providing the preprocessed reference in a convenient format, together with some (R?) code to run the comparison (e.g. to generate plots similar to those of Fig. 6b). Alternatively, tools based on reference count matrices, like scmap (PMID: 29608555), could be used. I would include some information in this sense to guide the readers in this kind of analysis.*

Response: We have uploaded a Table (GSE126030) to GEO containing the preprocessed data matrix of molecular counts for the cells profiled in our study. For convenience, we have included the cell-identifying barcodes for cells designated as T cells (see Methods) as part of the source data file for Fig. 6 where we present the complete reference map and its annotations (referred to on pg. 35). Furthermore, we have updated the text (pg. 35) to include the code for marker selection, clustering, and differential expression analysis (available at https://github.com/simslab/cluster_diffex2018, along with a tutorial detailing its use). Similarly, the code and tutorial for scHPF is available at <https://github.com/simslab/scHPF>. Lastly, the code for projection is now available at: https://github.com/simslab/umap_projection. Together with the available code and reference count matrices, readers will be able to more easily reconstruct our data and utilize the reference map for future studies.

METHOD ROBUSTNESS

3) *The authors present the bioinformatics methods adopted as “innovative” (l. 87), which I agree with. However, the field of scRNA-seq data analysis is fast-paced and far from being standardized. Many methods are available for the same analytical task, rely heavily on parameter setting and data preprocessing, and often return different results. Thus, the robustness of methods and results has to be endorsed by using orthogonal methods. For instance, I would suggest considering these methods in parallel with the analyses already performed:*

- *Projection of a query scRNA-seq dataset on a reference dataset with scmap (PMID: 29608555);*
- *Integration of different datasets with Seurat 3 (PMID: 31178118, see the description about the identification of “anchor” cells);*
- *Single-cell classification with SingleR (PMID: 30643263).*

These methods are also useful to confirm the projections of Fig. 2 and 6.

Finally, approaches like those implemented Seurat might be useful to integrate and harmonize several T-cell datasets to really provide an ultimate reference map. This is probably out of the scope of the current manuscript, but still worth mentioning in the Discussion section as a future development.

Response: We agree that validating computational results for the projections in Fig. 2 and 6 using orthogonal bioinformatic methods would strengthen our conclusions. As suggested, we repeated these analyses using *scmap*⁹, yielding projections that were both qualitatively and quantitatively consistent with our original UMAP projections (pg. 8, 26 and Supplementary Fig. 4 for Fig 2; pg. 14, 34 and Supplementary Fig. 15 for Fig. 6). Projections in *scmap* were highly correlated with those generated by UMAP across all datasets. Consistent with our findings in Fig. 2, T cells from blood projected onto T cells from bone marrow (BM), indicating that the blood T cell transcriptional profile is most similar to T cells from BM. In tumor-associated T cells (Fig 6.), analysis with *scmap* recapitulated findings obtained using UMAP, including a lack of activated CD4⁺ T cells and high concentrations of Tregs and functionally activated CD8⁺T cells in T cells derived from all tumor sites (Supplementary Fig. 15). There are some differences in the projection patterns obtained with *scmap* compared to UMAP, due to the different algorithms; *scmap* identifies the 10 most closely matched cells in the reference dataset, while UMAP depicts similarities based on the distance of individual cells compared to all other cells. The most prominent difference between *scmap* and the UMAP-based method is that cells projected with *scmap* cannot fall outside the boundaries of the reference map, while in UMAP some cells project outside the reference map, based on quantitative differences between cells.

The reviewer mentions additional methods for identifying correspondences between cells as done in Figs. 2 and 6, including SingleR and Seurat v3, the latter published after our manuscript was submitted. SingleR is a tool for annotating scRNA-seq data based on reference data from purified populations such as bulk RNAseq data; however, our dataset derives from tissue T cells and T cells activated for specific times for which reference datasets are not available. Therefore, singleR does not appear to be directly applicable to address similar questions that we investigated in this study. Seurat v3⁶, uses a different approach for comparing datasets by combining reference and query datasets together, generating new maps and clusters, which could be an interesting extension of our findings outside the scope of the present study. The data projection approaches we implemented here allow one to keep the reference map embedding constant while different data sets are projected onto it, which simplifies the comparison and visualization of multiple data sets. We have included statements about how using Seurat and other new methods for integrating and harmonizing datasets^{5-7,10} will be an important area for future studies of human T cells in health and disease (pg. 18 and see response for comment #1 above).

CLARITY OF METHOD DESCRIPTION

5) *In the Methods section, the description of the cell-type identification and scHPF analyses should be improved to facilitate understanding and reproducibility. I understand that cell-type identification was performed in three major steps:*

1. *Clustering for each donor separately, all tissues together, expecting K=12 clusters.*
2. *Cell-type classification.*
3. *Further selection of cell-type specific genes to be shown in Fig. 1c.*

The Methods section should describe clearly how steps 2 and 3 were performed, so to allow reproducibility.

Regarding step 2, I point out that recent best practices (PMID: 31217225) suggest integrating automatic (like SingleR) and manual approaches (based on prior knowledge of marker genes) to annotate cell types in large datasets like this one.

Response: We have included a comprehensive methods section, detailing all aspects of the cell acquisition, isolation and phenotypic and functional profiling, single cell transcriptome profiling and the many different analysis approaches used for the data. As suggested, we have provided additional information in both the methods and figure legends to more clearly describe how cell types and subset-specific genes were identified from the scRNAseq data in the revised manuscript. Importantly, the clustering was performed using the Phenograph implementation of Louvain clustering which is unsupervised and does not require the user to pre-determine the number of clusters. The parameter $k=12$ refers to the number of nearest neighbor cells assigned to each cell in the k -nearest neighbors graph generated prior to clustering.

We have revised the legend to Fig. 1c to include the methods for how genes were curated for visualization (step 3 above) as follows:

“Heatmaps show z-scored average expression of curated T cell subset marker genes that had a fold change >2 and $p < 0.05$ by the binomial test for at least one cluster. Genes are ordered by the cluster in which they have the highest fold change.” (pg. 40)

In the methods, we added the following sentence to the description of clustering:

“Differentially expressed genes for cells in each cluster versus all other cells were determined using a binomial test¹¹ (Supplementary Table 5,6).” (pg. 25)

Finally, we have added the following statement to the Code Availability section:

“Code for marker selection, clustering, and differential expression is available at https://github.com/simslab/cluster_diffex2018.” (Methods, pg. 35)

With respect to cluster labeling in Fig. 1 (step 2 above), we annotated clusters based on activation condition (i.e., whether the cell derived from a resting or activated sample), CD4:CD8 expression ratio (pg. 25), and expression of established T cell marker genes denoting effector memory (TEM), tissue resident memory (TRM), terminally differentiated effector cells (TEMRA), and regulatory T cells (Treg), as indicated in the legend to Fig.1. These cluster classifications were not used in downstream analysis, and several clusters were not annotated as specific subsets but designated based on sample origin as “CD4 Rest” or “CD4 Act”.

Finally, we note that automatic approaches to subset annotation (e.g. SingleR) are not applicable to our data because appropriate reference data for T cells in healthy tissues does not yet exist (see above). In particular, many of our profiled T cells are in an activated state, and not readily amenable to cell type annotation from bulk transcriptomic data (see comment above).

6) Similarly, for scHPF analysis:

1. Identification of factors specific for each sample (i.e. tissue-donor) using scHPF.
2. Clustering of the factor-factor correlation matrix to identify major modules.
3. Annotation/interpretation of modules.

Regarding step 2, I think the terminology is a bit confusing: are “modules” and “models” the same thing? Also, the sentence about the seven models mentioned at l. 629-630 is probably too premature and confusing because, at this point, only factors are identified, not modules. Step 3 requires a careful description of how the modules were annotated and why two modules could not be annotated. If possible, module-specific genes should be reported also for these two modules in Suppl. Table 7.

Response: To clarify, we have replaced the word “models” with “factorizations” as shown below and in the methods (pg. 29-30):

“scHPF was run with default parameters for seven values of K, the number of factors, equal to all values between 6-12 inclusively. This resulted in seven candidate scHPF *factorizations* per merged dataset”.

Our objective with this analysis was to identify gene signatures that were highly conserved across tissues and donors. As described in the Methods (pg. 30), “we focused on seven modules (out of nine) whose factors had a mean pairwise correlations greater than 0.25”. We did not examine the two modules that failed to meet this threshold because their factors were not well correlated with each other, and thus it was not clear that they represented coherent patterns that were common across donors and tissue. As we now describe in the main text (pg. 10), “modules were annotated based on known markers among their highest scoring genes, association with resting or activated states, and CD4:CD8 ratio”.

7) Please, also double check that the gene sets identified and used in Fig. 1c (differentiating T cell lineages), Fig. 3a-c (DEGs), and Fig. 3e (tissue signature) are fully reported in the Supplementary Material and referenced in the main text, and that the exact procedures for their identification is described in a Methods sub-section. The underlying methods and final gene-set results are not straightforward to picture out from the current manuscript.

Response: P-values and effect sizes for all genes in all clusters, some of which were chosen for visualization in Fig. 1c are reported in Supplementary Tables 5 & 6. As described above, we have added the following sentence in the methods to clarify:

“Differentially expressed genes for cells in each cluster versus all other cells were determined using a binomial test¹¹ (Supplementary Table 5,6).” (pg. 25)

Additionally, the specific genes visualized in Fig. 1c and all data for Fig. 3a-e and their associated expression values are reported in the Source Data files for those figures (now explicitly stated for each figure in the legend). The procedures by which they are generated are detailed in the methods, under the heading “Coarse-grained clustering of T cells from each donor” for Fig. 1c, and the heading “Comparison of TEM cells from tissue and blood” for Fig. 3a-e.

MAJOR POINTS:

8) I do not agree with the following sentence (l. 113-115): “T cells from BM and LN co-clustered while LG T cells were more distinct (Fig. 1b), consistent with T cell subset composition differences in these sites from phenotype analysis”.UMAP plots show clustering of cells based on similarities of the transcriptional profiles of single-cells, whereas sample composition is an

overall quantification of the different cell subtypes. These two aspects do not have a 1:1 correspondence but rather provide complementary insights. Also, the cellular composition is not easy to visualize from overlapping UMAP points. I would suggest rephrasing, as well as integrating both types of information in these plots (e.g. Fig. 6b), by showing both the UMAP plot and, next to it, a small, stacked barplot representing the cellular composition. In addition, it should be mentioned in the text that the quantification of cell subtypes from scRNA-seq data might be biased by single-cell dissociation efficiencies (PMID: 30345255).

Response: We have rephrased the text:

“T cells from BM and LN co-localized while LG T cells were more distinct (Fig. 1b), consistent with the greater proportion of CD8⁺T cells and TRM phenotype cells in LG relative to the two lymphoid sites (Supplementary Fig. 2).” (pg. 6)

For visual clarity, we have included pie charts of tissue T cell subset composition and stacked bar charts denoted tissue-resident memory phenotypes in Supplementary Fig. 2. We have also altered the text (pg. 18) to note that the quantification of cell types in our study may also be subject to dissociation biases between the individual tissues^{2,12}.

9) Being this work mainly based on dry- rather than wet-lab work, I am wondering whether you can confirm the results of your “kinetic analysis” of CD4⁺ T cells (p. 12) also computationally, for instance through analysis of CD4⁺ T cell “velocity” (PMID: 30089906) and evolution in pseudotime of the expression of the IFIT3 and NME1 genes (PMID: 24658644).

Response: Our conclusions regarding the temporal ordering of gene expression following T cell activation are based on computational (“dry lab”) inferences, and validated by experimental (“wetlab”) findings. Specifically, the identification of IFIT3 and NME1 as having distinct kinetics of expression following T cell activation was suggested from activation trajectories of the scRNAseq data generated using diffusion component analysis (Fig. 4), giving a pseudotemporal ordering of the IFN response module (marked by IFIT3 expression) relative to the Proliferation module (marked by NME1 expression) shown in Fig. 4d. This analysis is based on trajectory inference with diffusion component analysis¹³. We then sought to validate this computationally inferred ordering, using an experimental time course of human T cell activation, quantitating gene expression kinetics by qPCR as shown in Fig. 5. The time course confirmed that IFIT3 expression is upregulated early after T cell activation, reaches peak levels by 16hrs and declines thereafter (24-48hrs), while NME1 exhibits sustained upregulation 8-72hrs after activation. RNA velocity analysis is another computational approach to infer the directionality of cell state transitions for each cell; however, the realtime experimental results directly demonstrate the gene expression kinetics.

10) The transcriptional profiles of tumor-infiltrating CD8⁺ T cell were found to be consistent across all tissues, whereas the CD4⁺ T cell phenotypes were more variable. Could this result be due to a bias in the set of highly variable genes considered, as many of them were CD4-T-cell-specific, and much fewer CD8-T-cell-specific (as shown in Fig. 1c heatmap)?

Response: The heatmap in Fig. 1c shows a curated list of marker genes that were only used in the visualization for this figure, but were not used as highly variable genes for clustering (or for any analysis in the paper). The 315 highly variable genes used for projections and clustering are

listed in Supplementary Table 4 and have good representation of genes associated with both subsets.

11) Some interesting and original results might be put in a better light by discussing them more deeply. For instance, is there anything in the literature supporting the association found in this study between localization of T cells and structural changes?

Response: Our scRNAseq analysis reveals that tissue T cells exhibit changes in expression of key cytoskeletal and structural genes, which is a novel finding of this study. Previous investigations of human tissue T cells by our own laboratory and others have focused on identifying transcriptional and phenotypic profiles of tissue resident memory T cells¹⁴⁻¹⁷ or their adaptations to specific tissue sites^{18,19} using population RNA-seq. These previous studies have identified specific cell surface markers and functional profiles associated with TRM compared to circulating T cells in tissues and blood. In the present study, we query the individual transcriptomes of all T cells found within specific tissues compared to blood, revealing both TRM-associated genes and additional highly differentially expressed genes involved in cellular structure and cell-matrix interactions. Whether these changes are a common functional adaptation required for T cells entering and/or residing in tissue will be an important avenue for further research. We have modified the text in the discussion to further highlight these points (pg. 16).

12) Also, Treg transcriptional fingerprints have been previously reported to be very plastic and variable across tissues, and in your dataset seem to also differ between donors (Fig. 1b-c, resting in D1, activated in D2). Some discussion about the Treg phenotypes found in the different tissues of the present study would be extremely interesting for the scientific community.

Response: We agree that differences in Treg transcriptional profiles is an important research direction for future work. However, due to the very small numbers of Tregs in some tissues, there is not sufficient power to address this issue in the current study.

13) Besides Tregs, there is a lot of debate about the variability of immune-cell transcriptional profiles in different tissue and disease contexts, and on whether these differences should be taken into account by deconvolution methods for the quantification of cell subtypes from bulk transcriptomic data (PMID: 29541787). As you identified both tissue-dependent and -independent markers, a discussion of these aspects might provide valuable guidance for the researchers in this field.

Response: We agree that single-cell RNA-seq profiles and markers are an invaluable resource for deconvolution of bulk RNA-seq, which requires cell type-specific profiles or marker genes as input. While some of the markers we identified for tissues and activation states could be useful in this context, many of the markers are not necessarily specific for T cells. While our dataset may provide candidate markers for deconvolution of bulk mixtures of T cells to identify tissue-derived subsets or functional states, more validation is needed.

14) I agree that the current study can provide a basis to study T-cell phenotypes in cancer, but I find the analysis proposed too limited and not really conclusive. Thus, I would suggest

investigating a bit more its relevance for cancer immunology. One idea could be, for example, to test the prognostic and predictive value of the selected CD4-T-cell-activation markers using data from TCGA or from patients treated with immune checkpoint blockers, respectively.

Response: We demonstrate how our reference dataset can be applied to analyzing TILS from 4 different cancers profiled by different labs using different technologies. Importantly, our comparison showed highly consistent results across the four studies—that TILS contained functional effector CD8+ T cells and Tregs but lacked activated CD4+ T cells. These consistent findings indicate the robustness of our reference dataset. We further show that canonical markers of T cell exhaustion –a functional state associated with TILs, were also upregulated by activated T cells derived from healthy tissues. These results reveal the importance of obtaining baseline healthy profiles for high resolution analysis of T cells in disease on the single cell level, as we also emphasize in the discussion (p. 18). Moreover, researchers focused on cancer immunology can use and apply our findings to more large-scale assessments of T cells in different cancers, multiple patients and in the presence of immunotherapies. Such analyses are well beyond the scope of our study.

MINOR POINTS:

15) “CD3+T” (and similar ones) should be corrected to “CD3+ T” (with spaces after “+”).

Response: We have adjusted the text as suggested.

16) Please provide some rationale for the parameter setting presented in the Methods section (e.g. Louvain clustering with K=12 or DMAPS kernel bandwidth of 4).

Response: In Louvain clustering, not every cell is assigned to a cluster, and so it is important to choose parameters that minimize the number of unassigned cells. We chose K=12 because it was the highest resolution we could achieve without having a very large number of cells unassigned. The DMAPS kernel bandwidth was selected based on the heuristic suggested in Coifman et al²⁰ which states that the kernel bandwidth epsilon should fall within the linear range of a plot of a logscale plot of the sum over the Laplacian matrix elements (W_{ij}) vs. epsilon. We chose epsilon = 4 because it is the minimum value of epsilon in the linear range of the average plot for this data set, which we reasoned would give us the lowest noise trajectory without violating the heuristic as shown below:

More importantly, the overall shape of the diffusion trajectories are relatively insensitive to the choice of epsilon for values within the linear range of the above plot ($\text{eps} > 4$) as shown here for CD4+ T cells from the lung of Donor 1:

17) I would put the heatmaps of Fig. 1c in a separated panel (Fig. 1d) and use the same ordering of genes for the two heatmaps so to ease comparison.

Response: In Fig. 1c, the left side shows the UMAP projection of individual clusters for each tissue donor delineating distinct cell populations as determined by k-nearest neighbor analysis with each cluster is assigned an arbitrary number. The key differentially expressed genes which define each cluster are shown in the heat map on the right. The genes shown in each heatmap are a curated set of differentially expressed markers ordered by the cluster in which they have their maximum enrichment, as now indicated in the legend (p. 42). While the cluster numbers and gene lists generally agree for both donors, there are some differences in the gene rankings which are computationally determined.

18) In the second plot of Fig. 3f, the palette does not allow distinguishing well the groups.

Response: We have adjusted the colors to highlight the differences between the groups.

19) For the reasons explained above, I would not call the tissue gene signature “universal” (l. 148).

Response: We have revised the heading on pg.8 to “identification of a tissue gene signature common to multiple sites”.

20) Fig. 3a-c and the relative captions are a bit confusing. How were the red genes selected? The “several highlighted in red” explanation does not help in this sense. I would rather show the differentially expressed genes selected, together with the lines identifying the thresholds specified in the methods: $p\text{-adj} < 0.05$, $\log FC > 2$ (l. 573).

Response: The selected genes highlighted in red are among many differentially expressed genes (threshold $p\text{-adj} < 0.05$, fold change > 2) listed in the Source Data (tabs: Fig3a, Fig3b, and Fig3c). Given the number of significant genes, only some genes could be highlighted. We have mentioned this point in the figure legend (pg. 41).

21) What is exactly the Source Data (l. 961-962)? What do the clusters (1-10) on the columns of this file correspond to?

Response: The Source Data file contains raw data underlying all figures in the manuscript, as requested by the journal. Each tab corresponds to an individual figure panel. The clusters (Donor 1 clusters 1-11, Donor 2 clusters 1-10) in the first two tabs denote the gene expression values for each cluster in Fig. 1c for both donors.

22) Please, revise the following sentence (l.953-956) to better clarify what is presented in Fig. 3d and what is the meaning of “certain genes”: “Heatmap shows z-scored average expression for all genes in the tissue signature from (d) among the resting CCL5+ 954 T cells from each tissue and blood sample plus that of the rare blood subpopulation from (d), which expresses high levels of certain genes”.

Response: We have altered the text to read “...which express high levels of a subset of tissue signature genes”.

23) *I would add x- and y-axis labels to Suppl. Fig. 7a.*

Response: The clustergram in Supplementary Fig. 7a (now Supplementary Fig. 8a) is a symmetric matrix of correlation coefficients. The clusters within the x-axis which are labeled with a functional module designation are the same as in the y axis.

24) *Where is it explained in the Methods how were the diffusion maps built?*

Response: We described how diffusion maps were built under the heading “Activation Trajectory Analysis” in the method section. For clarity, we have changed the heading to “Activation Trajectory/Diffusion Analysis” (pg. 30).

25) *I would add the number of samples/replicates to Fig. 5 panels.*

Response: The number of biological replicates for all panels in Fig. 5 are stated in figure legends as per journal guidelines.

26) *What is the intended meaning of “biased” in the sentence at l. 261?*

Response: Our intention was to highlight that the induction and transient nature of IFIT3 expression was more evident in CD4⁺ T cells relative to CD8⁺ T cells. We have removed the word “biased” to avoid confusion.

27) *How were exhaustion markers quantified as a single variable in Fig. 7?*

Response: As described in the Fig. 7 caption, cells are colored by “the average expression of a set of exhaustion markers (PDCD1, CTLA4, LAYN, LAG3, TIM-3, CD244, and CD160)”.

28) *The association between a subset of CD8+ T cells with proliferation markers discussed at l. 312-314 would be probably more easily measured and visualized by using the “proliferation module” previously identified, instead of the single markers (which might also be sensitive to dropout).*

Response: Our T cell stimulation was relatively short (16h). As a result, the proliferation module reflects the earliest stages of cell cycle entry (e.g. G0S2 which is a marker of the transition from G0 to G1). In contrast, Ki67 is an established marker of actively dividing cells across multiple stages of the cell cycle. Our interpretation is that the KI67 positive tumor-associated T cells are at a more advanced stage of proliferation than those marked by our proliferation module.

29) *What is the intended meaning of “relationship” at l. 333? Are you referring to transcriptional/functional differences between the two types of TEM? Please, clarify.*

Response: Our intention was to refer to transcriptional differences between tissue-localized and circulating TEM subsets and have modified the text accordingly (pg. 16).

30) Throughout the manuscript, I would suggest referring to “transcriptional profiles” rather than “functional states” of T cells, as the actual functional states were not assessed in the study, but only the transcriptomes that could underlie them.

Response: Our study identifies transcriptional profiles in the context of a functional response comparing TCR-activated T cells to unstimulated controls. Thus, these profiles reflect functional states and changes at the level of transcription. Additionally, there are well established functional states of T cells, which we used to annotate the gene signatures we identified. For example, the cytotoxic module contains several genes encoding for well-established functional markers of cytotoxicity such as perforin and different granzymes. Likewise, the interferon response module is comprised of well-established IFN-inducible genes, which require IFN γ signaling for their induction.

31) What is the rationale behind the selection of models with $p_{adj} \geq 0.05$ (l. 636)?

Response: We wanted to choose a number of factors such that there was not a significant overlap between factors' top genes. As $p < 0.05$ is a standard threshold for significance, we chose the largest number of factors such that overlap was not significant at a $p < 0.05$ level (ie. $p \geq 0.05$). We note that a stronger significance threshold like 0.01 would have actually been more permissive because we are interested in a factorization without significant overlap.

References

1. van den Brink, S.C., Sage, F., Vertesy, A., Spanjaard, B., Peterson-Maduro, J., Baron, C.S., Robin, C. & van Oudenaarden, A. Single-cell sequencing reveals dissociation-induced gene expression in tissue subpopulations. *Nat Methods* **14**, 935-936 (2017).
2. Finotello, F. & Eduati, F. Multi-Omics Profiling of the Tumor Microenvironment: Paving the Way to Precision Immuno-Oncology. *Front Oncol* **8**, 430 (2018).
3. Regev, A., Teichmann, S.A., Lander, E.S., Amit, I., Benoist, C., Birney, E., Bodenmiller, B., Campbell, P., Carninci, P., Clatworthy, M., Clevers, H., Deplancke, B., Dunham, I., Eberwine, J., Eils, R., Enard, W., Farmer, A., Fugger, L., Gottgens, B., Hacohen, N., Haniffa, M., Hemberg, M., Kim, S., Klenerman, P., Kriegstein, A., Lein, E., Linnarsson, S., Lundberg, E., Lundberg, J., Majumder, P., Marioni, J.C., Merad, M., Mhlanga, M., Nawijn, M., Netea, M., Nolan, G., Pe'er, D., Phillipakis, A., Ponting, C.P., Quake, S., Reik, W., Rozenblatt-Rosen, O., Sanes, J., Satija, R., Schumacher, T.N., Shalek, A., Shapiro, E., Sharma, P., Shin, J.W., Stegle, O., Stratton, M., Stubbington, M.J.T., Theis, F.J., Uhlen, M., van Oudenaarden, A., Wagner, A., Watt, F., Weissman, J., Wold, B., Xavier, R., Yosef, N. & Human Cell Atlas Meeting, P. The Human Cell Atlas. *Elife* **6**(2017).
4. Lopez, R., Regier, J., Cole, M.B., Jordan, M.I. & Yosef, N. Deep generative modeling for single-cell transcriptomics. *Nat Methods* **15**, 1053-1058 (2018).
5. Haghverdi, L., Lun, A.T.L., Morgan, M.D. & Marioni, J.C. Batch effects in single-cell RNA-sequencing data are corrected by matching mutual nearest neighbors. *Nat Biotechnol* **36**, 421-427 (2018).
6. Stuart, T., Butler, A., Hoffman, P., Hafemeister, C., Papalexi, E., Mauck, W.M., 3rd, Hao, Y., Stoekius, M., Smibert, P. & Satija, R. Comprehensive Integration of Single-Cell Data. *Cell* **177**, 1888-1902 e1821 (2019).
7. Barkas, N., Petukhov, V., Nikolaeva, D., Lozinsky, Y., Demharter, S., Khodosevich, K. & Kharchenko, P.V. Joint analysis of heterogeneous single-cell RNA-seq dataset collections. *Nat Methods* **16**, 695-698 (2019).
8. Hie, B., Bryson, B. & Berger, B. Efficient integration of heterogeneous single-cell transcriptomes using Scanorama. *Nat Biotechnol* **37**, 685-691 (2019).
9. Kiselev, V.Y., Yiu, A. & Hemberg, M. scmap: projection of single-cell RNA-seq data across data sets. *Nat Methods* **15**, 359-362 (2018).
10. Butler, A., Hoffman, P., Smibert, P., Papalexi, E. & Satija, R. Integrating single-cell transcriptomic data across different conditions, technologies, and species. *Nat Biotechnol* **36**, 411-420 (2018).
11. Shekhar, K., Lapan, S.W., Whitney, I.E., Tran, N.M., Macosko, E.Z., Kowalczyk, M., Adiconis, X., Levin, J.Z., Nemesh, J., Goldman, M., McCarroll, S.A., Cepko, C.L., Regev, A. & Sanes, J.R. Comprehensive Classification of Retinal Bipolar Neurons by Single-Cell Transcriptomics. *Cell* **166**, 1308-1323 e1330 (2016).
12. Lambrechts, D., Wauters, E., Boeckx, B., Aibar, S., Nittner, D., Burton, O., Bassez, A., Decaluwe, H., Pircher, A., Van den Eynde, K., Weynand, B., Verbeken, E., De Leyn, P., Liston, A., Vansteenkiste, J., Carmeliet, P., Aerts, S. & Thienpont, B. Phenotype molding of stromal cells in the lung tumor microenvironment. *Nat Med* **24**, 1277-1289 (2018).
13. Haghverdi, L., Buettner, F. & Theis, F.J. Diffusion maps for high-dimensional single-cell analysis of differentiation data. *Bioinformatics* **31**, 2989-2998 (2015).
14. Kumar, B.V., Ma, W., Miron, M., Granot, T., Guyer, R.S., Carpenter, D.J., Senda, T., Sun, X., Ho, S.H., Lerner, H., Friedman, A.L., Shen, Y. & Farber, D.L. Human Tissue-Resident Memory T Cells Are Defined by Core Transcriptional and Functional Signatures in Lymphoid and Mucosal Sites. *Cell Rep* **20**, 2921-2934 (2017).
15. Watanabe, R., Gehad, A., Yang, C., Scott, L.L., Teague, J.E., Schlapbach, C., Elco, C.P., Huang, V., Matos, T.R., Kupper, T.S. & Clark, R.A. Human skin is protected by four functionally and

- phenotypically discrete populations of resident and recirculating memory T cells. *Sci Transl Med* **7**, 279ra239 (2015).
16. Hombrink, P., Helbig, C., Backer, R.A., Piet, B., Oja, A.E., Stark, R., Brassler, G., Jongejan, A., Jonkers, R.E., Nota, B., Basak, O., Clevers, H.C., Moerland, P.D., Amsen, D. & van Lier, R.A. Programs for the persistence, vigilance and control of human CD8+ lung-resident memory T cells. *Nat Immunol* **17**, 1467-1478 (2016).
 17. Pallett, L.J., Davies, J., Colbeck, E.J., Robertson, F., Hansi, N., Easom, N.J.W., Burton, A.R., Stegmann, K.A., Schurich, A., Swadling, L., Gill, U.S., Male, V., Luong, T., Gander, A., Davidson, B.R., Kennedy, P.T.F. & Maini, M.K. IL-2high tissue-resident T cells in the human liver: Sentinels for hepatotropic infection. *J Exp Med* **214**, 1567-1580 (2017).
 18. Miron, M., Kumar, B.V., Meng, W., Granot, T., Carpenter, D.J., Senda, T., Chen, D., Rosenfeld, A.M., Zhang, B., Lerner, H., Friedman, A.L., Hershberg, U., Shen, Y., Rahman, A., Luning Prak, E.T. & Farber, D.L. Human Lymph Nodes Maintain TCF-1(hi) Memory T Cells with High Functional Potential and Clonal Diversity throughout Life. *J Immunol* **201**, 2132-2140 (2018).
 19. Cheuk, S., Schlums, H., Gallais Serezal, I., Martini, E., Chiang, S.C., Marquardt, N., Gibbs, A., Detlofsson, E., Introini, A., Forkel, M., Hoog, C., Tjernlund, A., Michaelsson, J., Folkersen, L., Mjosberg, J., Blomqvist, L., Ehrstrom, M., Stahle, M., Bryceson, Y.T. & Eidsmo, L. CD49a Expression Defines Tissue-Resident CD8+ T Cells Poised for Cytotoxic Function in Human Skin. *Immunity* **46**, 287-300 (2017).
 20. Coifman, R.R., Shkolnisky, Y., Sigworth, F.J. & Singer, A. Graph Laplacian tomography from unknown random projections. *IEEE Trans Image Process* **17**, 1891-1899 (2008).

REVIEWERS' COMMENTS:

Reviewer #4 (Remarks to the Author):

The authors have satisfactorily addressed my major concerns and I recommend the manuscript for publication.

Point by Point Response to Reviewer Comments
“Single-cell transcriptomics of human T cells reveals tissue and activation signatures in health and disease”

NCOMM-16425B

Corresponding Authors: Donna L. Farber and Peter A. Sims

We are submitting a revised version of our manuscript to comply with the journal’s formatting requirements. The reviewer was satisfied with the revisions made in the previous version.

REVIEWERS' COMMENTS:

The authors have satisfactorily addressed my major concerns and I recommend the manuscript for publication.

Response: We thank the reviewer for evaluating our revised manuscript.